# SMG5-SMG7 authorize nonsense-mediated mRNA decay by enabling SMG6 endonucleolytic activity

Volker Boehm [1,2,7✉], Sabrina Kueckelmann [1,2,7], Jennifer V. Gerbracht[1,2], Sebastian Kallabis[3], Thiago Britto-Borges [4,5], Janine Altmüller[6], Marcus Krüger [2,3], Christoph Dieterich[4,5] & Niels H. Gehring [1,2✉]

Eukaryotic gene expression is constantly controlled by the translation-coupled nonsense-mediated mRNA decay (NMD) pathway. Aberrant translation termination leads to NMD activation, resulting in phosphorylation of the central NMD factor UPF1 and robust clearance of NMD targets via two seemingly independent and redundant mRNA degradation branches. Here, we uncover that the loss of the first SMG5-SMG7-dependent pathway also inactivates the second SMG6-dependent branch, indicating an unexpected functional connection between the final NMD steps. Transcriptome-wide analyses of SMG5-SMG7-depleted cells confirm exhaustive NMD inhibition resulting in massive transcriptomic alterations. Intriguingly, we find that the functionally underestimated SMG5 can substitute the role of SMG7 and individually activate NMD. Furthermore, the presence of either SMG5 or SMG7 is sufficient to support SMG6-mediated endonucleolysis of NMD targets. Our data support an improved model for NMD execution that features two-factor authentication involving UPF1 phosphorylation and SMG5-SMG7 recruitment to access SMG6 activity.

[1] Institute for Genetics, University of Cologne, Cologne, Germany. [2] Center for Molecular Medicine Cologne (CMMC), University of Cologne, Cologne, Germany. [3] CECAD Research Center, University of Cologne, Cologne, Germany. [4] Section of Bioinformatics and Systems Cardiology, Department of Internal Medicine III and Klaus Tschira Institute for Integrative Computational Cardiology, Heidelberg University Hospital, Heidelberg, Germany. [5] DZHK (German Centre for Cardiovascular Research), Partner site Heidelberg/Mannheim, Heidelberg, Germany. [6] Cologne Center for Genomics (CCG), University of Cologne, Cologne, Germany. [7] These authors contributed equally: Volker Boehm, Sabrina Kueckelmann. ✉email: boehmv@uni-koeln.de; ngehring@uni-koeln.de

Error-free and precisely regulated gene expression is an essential prerequisite for all living organisms. In eukaryotes, transcription and translation are controlled and fine-tuned by diverse mechanisms to ensure the generation of flawless RNAs and proteins[1]. Mature mRNAs that have completed all co-transcriptional and post-transcriptional processing steps and passed the associated quality checks are translated into proteins as the final step of gene expression in the cytoplasm. At this point, translation-coupled mechanisms inspect the mRNA one last time to perform final quality control. Specifically, it is assessed whether the translated mRNAs are legitimate or contain features indicating that these transcripts encode non-functional, incomplete, or potentially harmful proteins and therefore have to be degraded[2,3]. The arguably most famous translation-coupled quality control process is nonsense-mediated mRNA decay (NMD), which is best known for its role to remove mutated transcripts containing a premature termination codon (PTC)[4]. However, the relevance of NMD for cellular maintenance goes beyond the quality control function and is not restricted to mutated transcripts[5]. Previous studies found that about 5–10% of the expressed genes are affected by NMD in different organisms[6–15], suggesting that NMD serves as a regulatory mechanism, which fine-tunes general gene expression and helps to minimize the production of aberrant transcript isoforms. Furthermore, defects in the core NMD machinery are not compatible with life in higher eukaryotes[16–22], underlining the importance of NMD to function properly during development and cellular maintenance.

In general, inefficient translation termination seems to be the primary stimulus for NMD initiation[4]. Recent evidence suggests that NMD can in principle be triggered by each translation termination event with a certain probability[23]. In higher eukaryotes, this probability can be modulated by different NMD-activating features, such as a long 3' untranslated region (UTR)[24–27]. However, the exact length and composition of an NMD-activating 3' UTR is not exactly defined and many mRNAs contain NMD-suppressing sequences that allow them to escape this type of NMD[28–30]. Another potent activator of NMD is the presence of an RNA-binding protein complex called the exon-junction complex (EJC) downstream of a terminating ribosome[24,31–37]. The EJC serves as a mark for successful splicing and is deposited onto the mRNA approximately 20–24 nucleotides upstream of spliced junctions[38–41]. Stop codons are typically located in the last exon of regular protein-coding transcripts, thus ribosomes usually displace all EJCs from a translated mRNA, effectively removing the degradation-inducing feature. However, mutations or alternative splicing may produce isoforms with stop codons situated upstream of EJC deposition sites. Translation of these transcripts would fail to remove all EJCs and subsequently triggers the decay of the mRNA via efficiently activated NMD.

Intensive research over several decades uncovered the central players of the complex NMD pathway and how they cooperate to achieve highly specific and efficient mRNA degradation. According to generally accepted models, NMD execution requires a network of factors to identify a given translation termination event as aberrant[42]. The RNA helicase UPF1 holds a central position in the NMD pathway, as it serves as a binding hub for other NMD factors and is functionally involved in all stages from the recognition of NMD substrates until the disassembly of the NMD machinery[43]. In translation-inhibited conditions, UPF1 has the potential to bind non-specifically to all expressed transcripts. However, in unperturbed cells, UPF1 is found preferentially in the 3' UTR region of translated mRNA due to the displacement from the 5' UTR and coding region by translating ribosomes[44–47]. Furthermore, the ATPase and helicase activity of UPF1 is required to achieve target discrimination, resulting in increased binding of NMD-targets and release of UPF1 from non-target mRNAs[48].

If the translated mRNA contains a premature or otherwise aberrant termination codon, the downstream bound UPF1 promotes the recruitment of NMD factors and their assembly into an NMD-activating complex. Subsequently, protein-protein interactions between UPF1, UPF2, UPF3B, and—if present—the EJC stimulates the phosphorylation of (S/T)Q motifs in UPF1 by the kinase SMG1[49–53]. Importantly, the activity of the kinase SMG1 is regulated by multiple accessory NMD factors, presumably to prevent unwanted UPF1 phosphorylation on non-NMD targets[54–58]. The continued presence of UPF1 on the target transcript in an NMD-activating environment leads to gradually increasing hyper-phosphorylation of UPF1 at up to 19 potential phosphorylation sites[59]. The progressively phosphorylated residues in the N-terminal and C-terminal tails of UPF1 then act as binding sites for the decay-inducing factors SMG5, SMG6, and SMG7[60]. In this basic model, hyper-phosphorylation of UPF1 represents a "point of no return" for NMD activation, which effectively sentences the mRNA for degradation[61].

The final execution of NMD is divided into two major branches. The first branch relies on the interaction of phosphorylated UPF1 with the heterodimer SMG5-SMG7, which in turn recruits the CCR4-NOT deadenylation complex[62–64]. Consequently, SMG5-SMG7 promote target mRNA deadenylation, followed by decapping and 5'–3' or 3'–5' exonucleolytic degradation[65,66]. The second branch is mediated by the endonuclease SMG6, which interacts with UPF1 to cleave the NMD-targeted transcript in a region around the NMD-activating stop codon[36,67–71]. This endonucleolytic cleavage results in the generation of two decay intermediates, which are rapidly removed by exonucleolytic decay. Both SMG5-SMG7 and SMG6-mediated degradation pathways are considered to be redundant, as they target the same transcripts[14]. They are also regarded as independent because downregulation of individual factors (SMG5, SMG6, or SMG7) only partially inhibits NMD[35,62,72]. However, loss of SMG6 impaired NMD more severely than inactivation of SMG7, suggesting that endonucleolytic cleavage is the preferred decay pathway, whereas deadenylation has merely a backup/supplementary function[14,35,62,70,71,73]. Nonetheless, the apparent redundancy hampered a detailed investigation of the final steps of NMD so far, since the inactivation of one decay route seemed to be partially compensated by the other.

Here, we addressed the central question if and how SMG5-SMG7 and SMG6 functionally cooperate and influence each other. We hypothesized that the inactivation of SMG5-SMG7 should activate the SMG6-dependent NMD pathway and still permit normal NMD if both pathways are independent. However, we show here that the combined loss of SMG5-SMG7 efficiently inactivates NMD and that SMG6 was catalytically inactive in cells depleted of SMG5-SMG7. This demonstrates that SMG6 is not independent of SMG5-SMG7 and could not compensate for their loss. This was especially surprising given that SMG6 was previously considered to be the dominant NMD-executing factor. Exploring the potential mechanism, we find that SMG7 requires the interaction with SMG5 and phosphorylated UPF1 for full NMD activity, whereas SMG5 supports NMD even in the absence of SMG7. We propose a model, in which either individual or combined SMG5 and SMG7 recruitment to hyper-phosphorylated UPF1 acts as an additional licensing step required for SMG6-mediated degradation of the target transcript. This model of two-factor authentication explains the tight control of NMD on regular transcripts, which prevents the untimely access of endonucleolytic decay activities.

## Results

**NMD is impaired in SMG7 knockout cells.** Phosphorylation of UPF1 represents a central checkpoint in NMD, which is followed

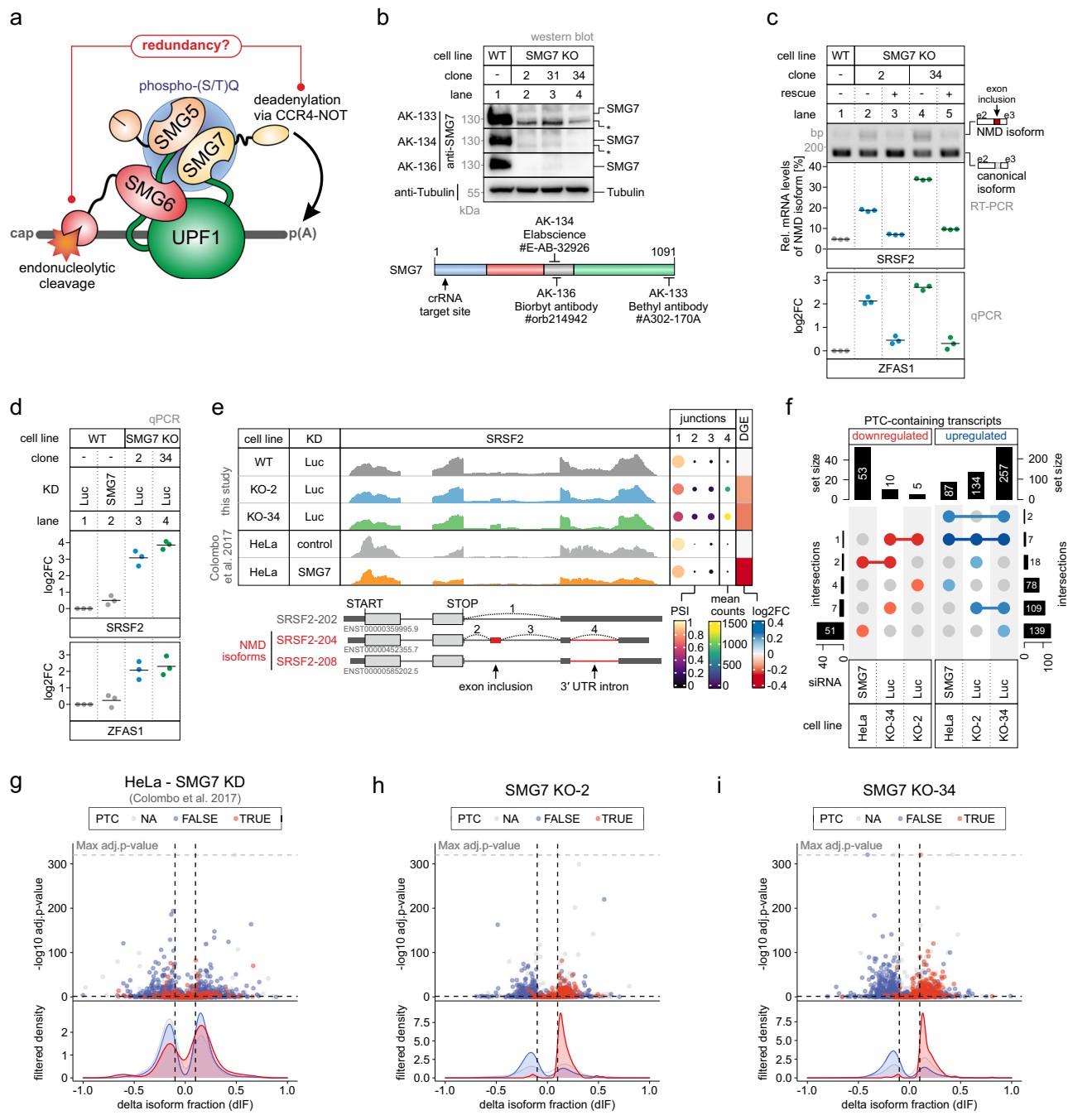

by SMG6-mediated endonucleolytic cleavage and/or SMG5-SMG7-mediated deadenylation of the target transcript (Fig. 1a). We hypothesized that after deactivating the deadenylation-dependent NMD branch, the execution of NMD would rely exclusively on the activity of SMG6. To achieve this goal, we generated SMG7 knockout (KO) Flp-In-T-REx-293 cells and identified three clones lacking the SMG7-specific band in western blot analyses using three different antibodies (Fig. 1b and Extended Data Fig. 1a, b). In all clones, the SMG7 genomic locus contained different frame-shift-inducing insertions/deletions, which also resulted in altered splicing of the CRISPR-targeted SMG7 exon. (Extended Data Fig. 1c–e). Phenotypically, the SMG7 KO clones proliferated slower compared to the wild-type (WT) cells, with no apparent decrease in cell survival (Extended

Data Fig. 1f). These results indicate that the depletion of full-length SMG7 protein impairs cellular fitness, presumably due to reduced NMD capacity. To test if NMD is indeed impaired in SMG7 KO cells, we quantified the levels of two exemplary endogenous NMD targets. SRSF2 is a serine/arginine-rich (SR) splicing factor, which auto-regulates its expression by generating NMD-sensitive splice isoforms of its mRNA[74]. ZFAS1 is a snoRNA host mRNA with a short PTC-containing ORF, which was reported as an NMD target undergoing SMG6-dependent endocleavage[70]. In the tested SMG7 KO clones (2 and 34) the levels of the NMD-sensitive SRSF2 isoforms and the ZFAS1 mRNA were markedly upregulated (Fig. 1c). Baseline levels of these NMD substrates were restored by expressing the WT SMG7 protein from genomically integrated constructs. Importantly, the

**Fig. 1 SMG7 depletion impairs NMD activity. a** Schematic representation of the final steps of nonsense-mediated mRNA decay (NMD). Phosphorylated UPF1 (indicated by the blue sphere) recruits the SMG5-SMG7 heterodimer to the target mRNA (indicated in dark gray), thereby promoting deadenylation via the CCR4-NOT complex. Recruitment of SMG6 to UPF1 results in endonucleolytic cleavage of the target transcript via the activity of the SMG6 PIN domain. The SMG5 PIN domain is catalytically inactive. **b** Western blot analysis of SMG7 knockout (KO) cell lines (clones 2, 31, and 34) with the anti-SMG7 antibodies AK-133, AK-134, and AK-136 (n = 1); Tubulin serves as control (see "Methods" section and Supplementary Data 6 for antibody details). The region of SMG7 detected by the antibodies is schematically depicted and the crRNA targeting site indicated. Asterisks indicate non-specific bands. **c** End-point RT-PCR detection of SRSF2 transcript isoforms (top) and quantitative RT-PCR-based detection (qPCR; bottom) of ZFAS1 in the indicated cell lines with or without expression of FLAG-tagged SMG7 as rescue construct. The detected SRSF2 isoforms are indicated on the right, the NMD-inducing included exon is marked in red (e = exon). Relative mRNA levels of SRSF2 isoforms were quantified from bands of agarose gels (n=3 biologically independent samples). The ratio of ZFAS1 to the C1orf43 reference was calculated; data points and means from the qPCRs are plotted as log2 fold change (log2FC) (n = 3 biologically independent samples). The plotted points are color-coded based on cell line (gray = WT; blue = SMG7 KO-2; green = SMG7 KO-34). **d** Quantitative RT-PCR-based detection (qPCR) of SRSF2 isoforms and ZFAS1 in the indicated cell lines upon treatment with the indicated siRNA. The ratio of NMD isoform to canonical isoform (SRSF2) and ZFAS1 to the C1orf43 reference was calculated; data points and means from the qPCRs are plotted as log2 fold change (log2FC) (n = 3 biologically independent samples). **e** Read coverage of SRSF2 from SMG7 KO and published SMG7 KD (GEO: GSE86148) RNA-Seq data is shown as Integrative Genomics Viewer (IGV) snapshots. The canonical and NMD-sensitive isoforms are schematically indicated below. Percent spliced in (PSI; from LeafCutter analysis) and mean counts from 4 indicative splice junctions are shown. Differential gene expression (from DESeq2) is depicted as log2 fold change (log2FC) in the last column. **f** Overlap of upregulated or downregulated premature termination codon (PTC)-containing isoforms between the SMG7 KO or KD RNA-Seq data is shown as UpSet plot. **g–i** Volcano plots showing the differential transcript usage (via IsoformSwitchAnalyzeR) in various SMG7 depletion RNA-Seq data. Isoforms containing GENCODE (release 33) annotated PTC (red, TRUE), regular stop codons (blue, FALSE), or having no annotated open reading frame (gray, NA) are indicated. The change in isoform fraction (dIF) is plotted against the −log10 adjusted p-value (adj.p-value). Density plots show the distribution of filtered isoforms in respect to the dIF, cutoffs were |dIF|> 0.1 and adj.p-value < 0.05. P-values were calculated by IsoformSwitchAnalyzeR using a DEXSeq-based test and corrected for multiple testing using the Benjamini-Hochberg method.

NMD defect was quantitatively more pronounced in the SMG7 KO cells compared to a siRNA-mediated knockdown (KD) of SMG7 in control cells (Fig. 1d).

To gain insights into the transcriptome-wide effects of the SMG7 depletion, we sequenced poly(A)+ enriched mRNA from both SMG7 KO clones and identified differentially expressed genes (Extended Data Fig. 2a and Supplementary Data 1). Consistent with the mRNA-degrading function of SMG7 in NMD, more than twice as many genes were upregulated than downregulated in the SMG7 KO cells (Extended Data Fig. 2b). Compared to a recently published study using SMG7 KD in HeLa cells[14], this ratio of upregulated vs. downregulated genes was higher (Extended Data Fig. 2b–e). We observed a substantial overlap between upregulated genes in both SMG7 KO cell lines, indicating that these genes are high-confidence SMG7 targets. In contrast, only a limited overlap between downregulated genes could be detected, suggesting these are rather clone-specific effects or off-targets (Extended Data Fig. 2b). From these analyses, we conclude that the KO of SMG7 leads to stronger NMD inhibition than the KD.

Next, we quantified alternative splicing events (Supplementary Data 2), as well as differential transcript usage, and identified significant isoform switches (Extended Data Fig. 2a and Supplementary Data 3). Isoform switches are characterized by significant changes in the relative contribution of isoforms to the overall gene expression when comparing two conditions[75]. Because of the identification at the isoform level, this approach allows the identification of PTC-containing transcripts that are upregulated upon NMD inhibition. As a specific example of such an isoform switch, we visualized the read coverage for the previously used bona fide NMD target SRSF2. While the overall SRSF2 expression remained nearly unchanged, we detected prominent NMD-inducing exon inclusion and 3′ UTR splicing events in the SMG7 KO but not in the SMG7 KD conditions (Fig. 1e). We verified the SMG7-dependent upregulation of additional examples by end-point PCR (Extended Data Fig. 2f–h). On a transcriptome-wide scale, NMD-sensitive isoforms with annotated PTC were almost exclusively detected to be upregulated in the SMG7 KO cells, which was not the case in the SMG7 KD in HeLa cells[14] (Fig. 1f–i). Collectively, the RNA-Seq data

analysis supported the initial observation that NMD is markedly impaired when SMG7 is knocked out. Due to the clear effect on NMD upon complete loss of SMG7, the KO cells provide an ideal background to examine further mechanistic details of NMD, which could not be studied before.

**SMG5 is required to maintain residual NMD in SMG7-depleted cells.** We utilized the SMG7 KO cells to investigate which domains and protein-protein interactions of SMG7 are required to support NMD. Specifically, we aimed to confirm whether the SMG5-SMG7 heterodimer initially binds phosphorylated UPF1 (p-UPF1) and subsequently triggers dead-enylation of the target mRNA via the recruitment of the CCR4-NOT complex (Fig. 1a). To this end, we generated stable SMG7 KO clone 2 cell lines that inducibly express SMG7 variants as rescue proteins. The 14-3-3$^{\text{mut}}$ was expected to be inactive in this assay due to the lack of interaction with UPF1 and p-UPF1 (Fig. 2a–c)[76]. However, the SMG7 14-3-3$^{\text{mut}}$ protein efficiently restored NMD activity in the SMG7 KO cells (Fig. 2d). This suggests that the p-UPF1 binding is not absolutely critical for the function of SMG7 in NMD. Next, we investigated if SMG7 has to form a heterodimer with SMG5 for full NMD activity. Surprisingly, the expression of a G100E mutant of SMG7 (unable to interact with SMG5; Fig. 2a–c)[62] failed to rescue the NMD defect (Fig. 2d). This finding was unexpected in the light of the currently advocated NMD model, in which SMG5 is merely a companion for SMG7 with the role to potentially strengthen the binding of the SMG5-SMG7 heterodimer to p-UPF1[62]. This finding prompted us to systematically address the question, which combinations of the three decay-inducing proteins SMG5, SMG6, and SMG7 are required for NMD (Fig. 2e and Extended Data Fig. 3a, b). Single SMG5 or SMG7 KDs in WT 293 cells resulted in very mild or nearly undetect-able inhibition of NMD, whereas depletion of SMG6 showed an intermediate effect, reflected by the upregulation of the SRSF2 NMD isoform and ZFAS1 (Fig. 2e and Extended Data Fig. 3b). Co-depletion of SMG6 and SMG5 via siRNAs showed a similar inhibitory effect on NMD as the single SMG6 KD (Fig. 2e and Extended Data Fig. 3b; lane 5 vs. lane 3). As expected, KD of SMG6 in the SMG7 KO cells strongly abolished NMD activity,

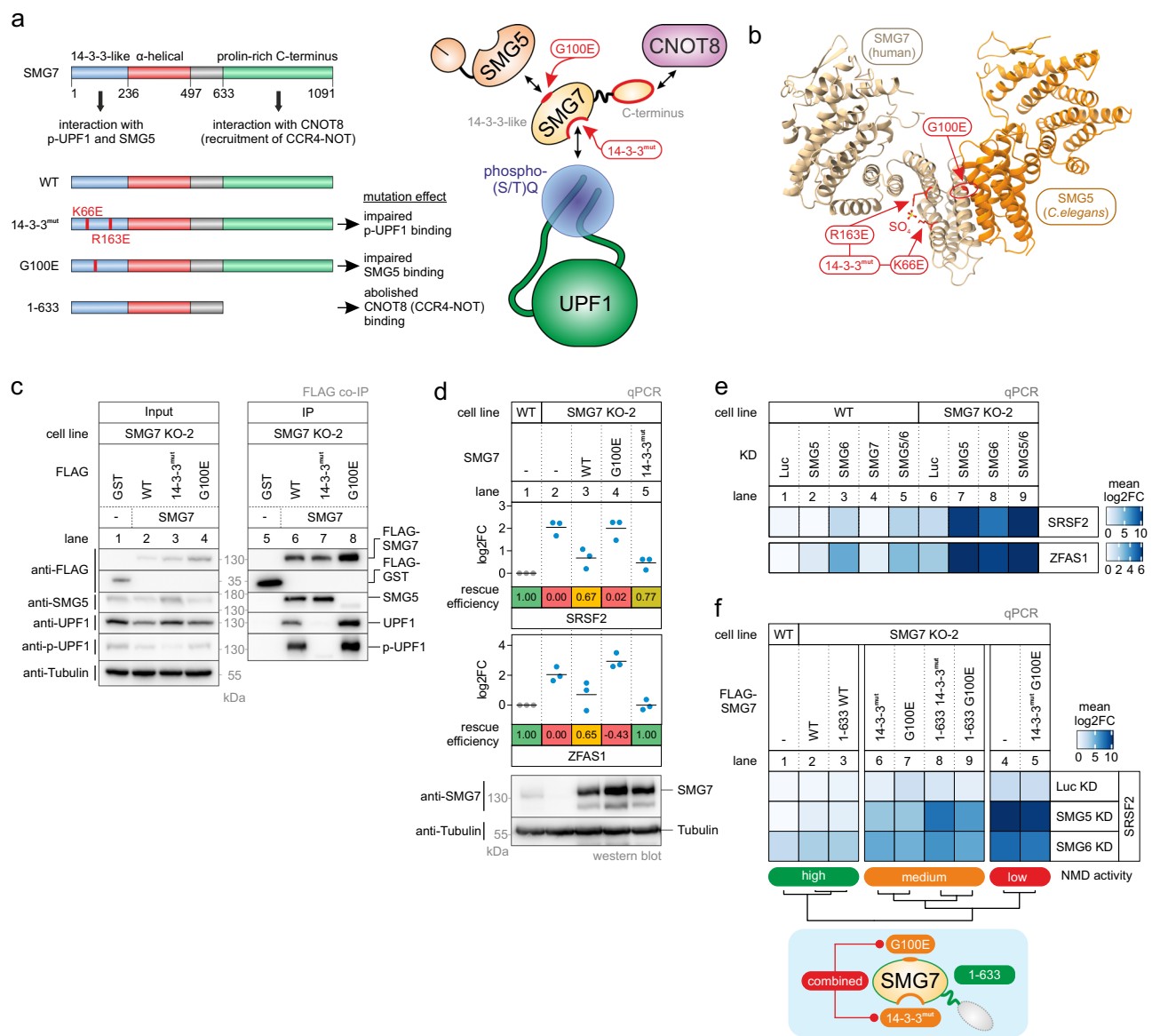

**Fig. 2 SMG7 requires interaction with SMG5 to rescue the SMG7 knockout phenotype. a** Schematic representation of the SMG7 domain structure on the left. The proposed functions of the domains are indicated and mutated constructs and their expected effect are shown below. The illustration on the right depicts which mutation is expected to impair which individual SMG7 function and/or protein-protein interaction. **b** Model of the SMG5-SMG7 heterodimer structure. Human SMG7 (PDB ID: 1YA0) was modeled on the *C. elegans* SMG5-SMG7 structure (PDB ID: 3ZHE). Critical SMG7 mutations are highlighted in red and the indicated sulfate ion mimics a phosphorylated residue. **c** Western blot after FLAG co-immunoprecipitation (IP) of FLAG-tagged GST (control) or SMG7 constructs in SMG7 KO cells ($n = 1$). Tubulin serves as a control. **d** Quantitative RT-PCR-based detection (qPCR) of SRSF2 isoforms and ZFAS1 was carried out in the indicated cell lines upon expression of the indicated FLAG-tagged rescue constructs. The ratio of NMD isoform to canonical isoform (SRSF2) and ZFAS1 to the C1orf43 reference was calculated; data points and means from the qPCRs are plotted as log2 fold change (log2FC) ($n = 3$ biologically independent samples). Rescue efficiency was calculated based on the mean log2FC in relation lane 1 (set to 1) and lane 2 (set to 0). Western blot analyses are shown below ($n = 1$). Tubulin serves as a control. **e** Heatmap of quantitative RT-PCR-based detection (qPCR) of SRSF2 isoforms and ZFAS1 in the indicated cell lines upon treatment with the indicated siRNA. The ratio of NMD isoform to canonical isoform (SRSF2) and ZFAS1 to the C1orf43 reference was calculated; mean log2 fold change (log2FC) is shown ($n = 3$ biologically independent samples). The corresponding individual data points are plotted in Extended Data Fig. 3b. **f** Heatmap of quantitative RT-PCR-based detection (qPCR) of SRSF2 isoforms in the indicated cell lines upon treatment with the indicated siRNA and expression of the indicated FLAG-tagged rescue constructs. The ratio of NMD isoform to canonical isoform (SRSF2) was calculated; mean log2 fold change (log2FC) is shown ($n = 3$ biologically independent samples). The corresponding individual data points are plotted in Extended Data Fig. 3c. Clustering ($k = 3$) and functional summary of SMG7 mutations for NMD activity are depicted below.

since both endonucleolytic and exonucleolytic pathways of NMD are inactivated (Fig. 2e and Extended Data Fig. 3b; lane 8). Remarkably and contrary to current NMD models, an even stronger NMD inhibition was observed when depleting SMG5 in the SMG7 KO cells, which was not further enhanced by the additional KD of SMG6 (Fig. 2e and Extended Data Fig. 3b; lane

9 vs. lane 7). This unanticipated result can explain our observation of why the SMG7 G100E mutant does not rescue the SMG7 KO and indicates that the SMG5-SMG7 heterodimer is critical for NMD, even when SMG6 is present. Moreover, these results expose a previously underestimated critical role of SMG5 in human NMD when SMG7 is impaired.

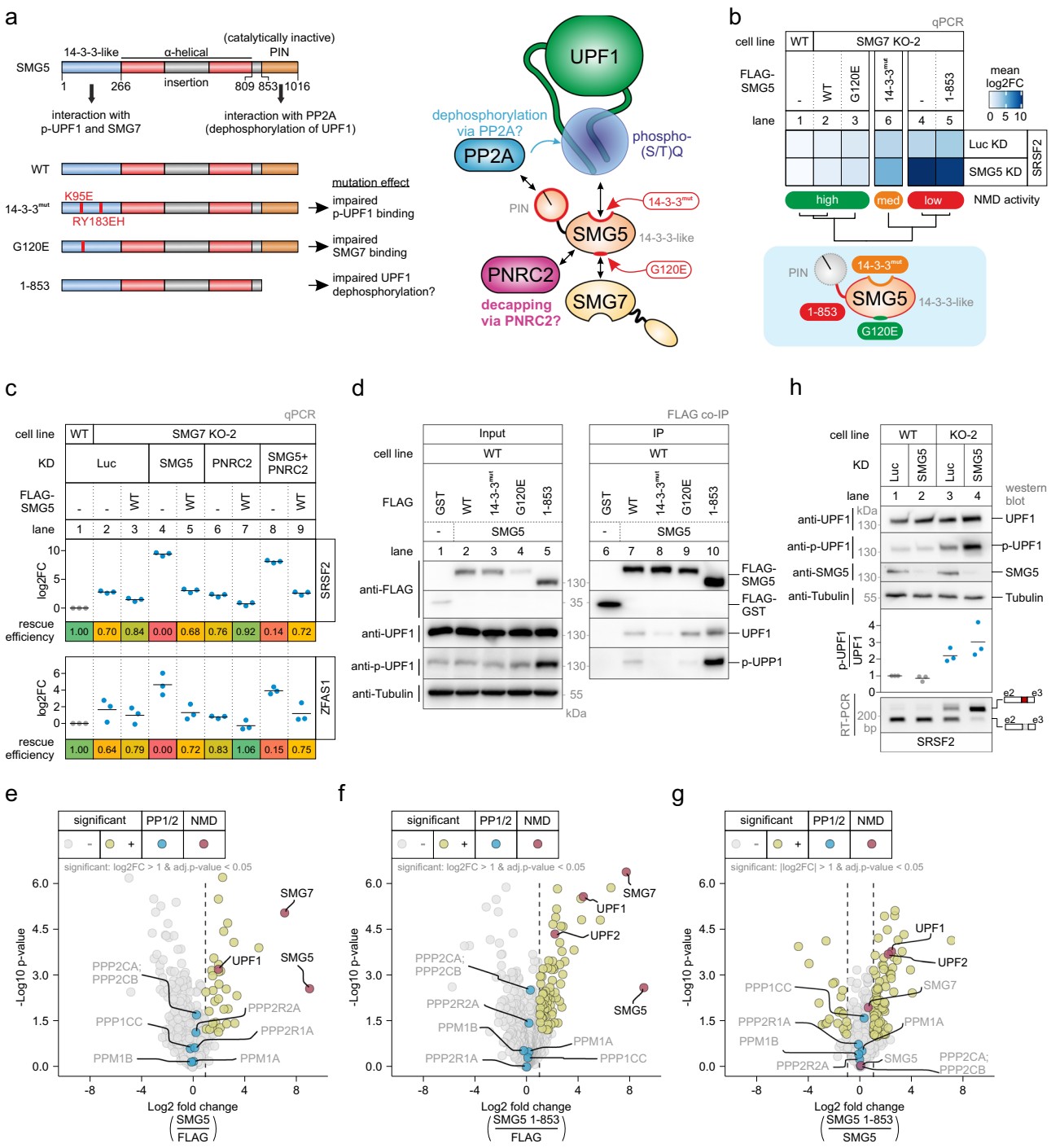

To gain further mechanistic insight into the cooperation between SMG5, SMG6, and SMG7, we performed complementation experiments in SMG7 KO cells with additional SMG7 rescue constructs and in combination with SMG5 or SMG6 KDs (Extended Data Fig. 3c). Three main clusters of rescue conditions were identified according to their NMD activity (Fig. 2f and Extended Data Fig. 3d). Consistent with our earlier observations, SMG7 WT rescued the SMG7 defect in all conditions and restored high NMD activity. The SMG7 1-633 deletion mutant (reported to be unable to recruit the CCR4-NOT complex; Fig. 2a)[64] also conferred high NMD activity in SMG7 KO cells, even when SMG5 or SMG6 were depleted in addition (Fig. 2f and Extended Data Fig. 3c, d). This result indicates that the

accelerated deadenylation of NMD substrates by SMG7-recruited CCR4-NOT is not required for their efficient degradation. In contrast to a previous report[64], the C-terminus of SMG7 is not required for NMD even when SMG6 is downregulated. However, by combining the C-terminal truncation with either 14-3-3[mut] or G100E mutations, these SMG7 variants became less NMD-competent, especially in the SMG5 KD condition (Fig. 2f and Extended Data Fig. 3c, d). This observation suggests that the deadenylation-inducing activity might serve as an additive but dispensable feature that helps to clear NMD targets, especially when other features of SMG7 are inactivated. In the full-length SMG7 context, the 14-3-3[mut] or G100E mutations displayed intermediate NMD rescue activity in the SMG5 and SMG6 KD

**Fig. 3 SMG5 expression rescues the SMG7 KO phenotype. a** Schematic representation of the SMG5 domain structure on the left. The proposed functions of the domains are indicated and mutated constructs and their expected effect are shown below. The illustration on the right depicts which mutation is expected to impair which individual SMG5 function and/or protein-protein interaction. **b** Heatmap of quantitative RT-PCR-based detection (qPCR) of SRSF2 isoforms in the indicated cell lines upon treatment with the indicated siRNA and expression of the indicated FLAG-tagged rescue constructs. The ratio of NMD isoform to canonical isoform (SRSF2) was calculated; mean log2 fold change (log2FC) is shown ($n = 3$ biologically independent samples). The corresponding individual data points are plotted in Extended Data Fig. 4a. Clustering ($k = 3$) and functional summary of SMG5 mutations for NMD activity are depicted below. **c** Quantitative RT-PCR-based detection (qPCR) of SRSF2 isoforms and ZFAS1 was carried out in the indicated cell lines upon treatment with the indicated siRNA and expression of the indicated FLAG-tagged rescue constructs. The ratio of NMD isoform to canonical isoform (SRSF2) and ZFAS1 to the C1orf43 reference was calculated; data points and means from the qPCRs are plotted as log2 fold change (log2FC) ($n = 3$ biologically independent samples). Rescue efficiency was calculated based on the mean log2FC in relation lane 1 (set to 1) and lane 4 (set to 0). **d** Western blot after FLAG co-immunoprecipitation (IP) of FLAG-tagged GST (control) or SMG5 constructs in WT cells ($n = 1$). Tubulin serves as a control. **e–g** Volcano plots of mass spectrometry-based analysis of the interaction partners of FLAG-tagged SMG5 WT or 1-853 constructs in WT cells ($n = 4$ biologically independent samples). **e** SMG5 WT against FLAG control, (**f**) SMG5 1-853 against FLAG control, (**g**) SMG5 1-853 against SMG5 WT. The yellow color labeling indicates targets that are significant in the respective comparisons after two-sided Welch's $t$-testing (log2 fold change (log2FC) >1 or |log2FC| >1; and adj. $p$-value <0.05). Points labeled in blue indicate phosphatase subunits of interest; points labeled in red indicate NMD factors. Highlighted proteins that were not significant in the respective comparisons are labeled with gray text. **h** Analysis of endogenous UPF1 serine 1127 (S1127) phosphorylation status in the indicated cell lines and knockdown conditions. Quantification results of phosphorylated UPF1 (p-UPF1) vs. total UPF1 are shown as data points and mean ($n = 3$ biologically independent samples). Knockdown of SMG5 protein as a western blot ($n = 2$ biologically independent samples) and the functional impact on SRSF2 isoform distribution (End-point RT-PCR as in Fig. 1c; $n = 3$ biologically independent samples) is shown.

conditions, whereas a 14-3-3$^{mut}$/G100E double-mutation was completely inactive in all conditions (Fig. 2f and Extended Data Fig. 3d). All KD/rescue experiments were also performed in the SMG7 KO clone 34 and showed similar functional outcomes (Extended Data Fig. 3e). In summary, these results reveal the synergistic effect of simultaneous binding of SMG7 to p-UPF1 and SMG5, whereas the lack of both interactions incapacitates the function of SMG7 in NMD.

Next, we performed co-immunoprecipitation (co-IP) experiments with the functionally tested set of FLAG-tagged SMG7 constructs to analyze their steady-state interaction with UPF1 (Extended Data Fig. 3f). The full-length or truncated 14-3-3$^{mut}$ SMG7 protein showed strongly impaired ability to co-immunoprecipitate UPF1, consistent with the role of the 14-3-3-like domain to mediate the stable interaction with phosphorylated UPF1 (Extended Data Fig. 3f; lane 10 vs. lanes 11 and 15)[63,77]. In turn, this result also indicates that SMG5 does not bridge the 14-3-3$^{mut}$ SMG7 protein to UPF1. The G100E mutants co-immunoprecipitated similar UPF1 levels than the respective WT construct (Extended Data Fig. 3f; lanes 10,14 vs. lanes 12,16), confirming that SMG7 does not require the heterodimerization with SMG5 to interact with UPF1[62]. In conjunction with the impaired functional rescue of the G100E mutants (Fig. 2f and Extended Data Fig. 3c–e), this result implies that disrupting the interaction of SMG7 with SMG5 impairs NMD even when the interaction between SMG7 and UPF1 is maintained. In conclusion, our functional studies reveal the previously underestimated requirement for SMG7 to interact with both p-UPF1 and SMG5 in order to support full NMD activity.

**SMG5 functionally complements the loss of SMG7.** The unexpected relevance of SMG5 prompted us to investigate the molecular properties of SMG5 in more detail. Similar to our analysis of SMG7, we aimed to understand which functions of SMG5 are required to maintain NMD competence in the presence or absence of SMG7. The first striking result was the almost complete rescue of the SMG7 depletion phenotype by the overexpression (~50-fold) of SMG5 WT or G120E mutant (reduced interaction with SMG7)[62] in control or SMG5 KD conditions (Fig. 3a, b and Extended Data Fig. 4a–c). This finding was very surprising since SMG5 is not expected to be able to directly carry out RNA degradation. Although SMG5 harbors a PilT N-terminus (PIN) domain that structurally resembles the functional PIN domain of SMG6, two of the three required catalytic residues

are missing in the inactive C-terminal SMG5 PIN domain (Fig. 3a)[78]. SMG5 was found in one study to promote degradation by interacting with the decapping factor PNRC2 (Fig. 3a), which could explain the NMD activity we observe in the rescue assays[79]. However, we did not observe any effects of PNRC2 KDs on the ability of SMG5 to rescue the NMD defects (Fig. 3c), consistent with other studies which failed to confirm this PNRC2-dependent path of SMG5-dependent degradation[64,65].

Collectively, the observation of SMG5 expression rescuing SMG7 loss once again calls into question the relevance of the SMG7 deadenylation-promoting function for NMD. We hypothesized that in the absence of SMG7, SMG5 interacts with p-UPF1 via its N-terminal 14-3-3-like domain to activate NMD. In line with this hypothesis, mutating three residues in the potential phosphopeptide binding pocket of SMG5 severely affected the ability of SMG5 to rescue the SMG7 KO phenotype in control or SMG5 KD conditions (Fig. 3b and Extended Data Fig. 4a, b). Of note, we could not detect any stable binding of SMG5 to UPF1 above background levels in SMG7 KO cells (Extended Data Fig. 4d), confirming that strong SMG5-UPF1 interactions require SMG7. However, in WT cells the 14-3-3$^{mut}$ SMG5 protein showed reduced binding to p-UPF1 (Fig. 3d).

We hypothesized that besides the interaction with UPF1 another function of SMG5 may be required to maintain NMD activity. To test this idea, we generated an SMG5 deletion mutant (1-853) lacking the C-terminal PIN domain (Fig. 3a). This SMG5 mutant, albeit being able to interact with UPF1 (Fig. 3d), was unable to restore NMD activity (Fig. 3b and Extended Data Fig. 4a, b), suggesting that the catalytically inactive PIN domain of SMG5 carries out an essential function during NMD. In search of an explanation, we considered earlier reports that the C-terminus of SMG5 is involved in the dephosphorylation of UPF1, likely by direct recruitment of the protein phosphatase 2 (PP2A) complex (Fig. 3a)[53,60]. Appropriately, we observed that the expression of the SMG5 1-853 construct leads to overall increased UPF1 phosphorylation, and consequently the SMG5 1-853 protein co-immunoprecipitated more p-UPF1 (Fig. 3d), indicating that the C-terminal PIN domain could play a role in the dephosphorylation of UPF1. To address whether the SMG5 PIN domain indeed enables the dephosphorylation of UPF1 via the direct recruitment of PP2A, we performed co-immunoprecipitation of FLAG-tagged SMG5 WT or 1-853 constructs in WT 293 cells, followed by label-free mass spectrometry (Supplementary Data 4). Although both SMG5 constructs

co-immunoprecipitated the direct binding partner SMG7 efficiently, we did not detect a significant enrichment of neither PP2A nor PP1 protein phosphatase complex components (Fig. 3e–g). We detected increased levels of UPF1 and UPF2 in the SMG5 1-853 co-immunoprecipitation compared to the SMG5 WT, whereas PNRC2 or other NMD components such as SMG6 were not enriched in any condition (Fig. 3e–g and Supplementary Data 4). Collectively, these findings do not provide evidence for but rather against the direct recruitment of PP2A via the SMG5 PIN domain, which questions the previously assumed role of SMG5 in UPF1 dephosphorylation. An alternative explanation could be that the SMG5 PIN domain is critical to promote NMD execution once SMG5 is bound to p-UPF1. Accordingly, the absence of the PIN domain would lead to the accumulation of NMD-incompetent UPF1 complexes. Supporting evidence for this hypothesis is the failed functional rescue of SMG5 1-853 (Fig. 3b and Extended Data Fig. 4a, b), the increased interaction with UPF1 and UPF2 (Fig. 3e–g), and the apparent hyper-phosphorylation of UPF1 (Fig. 3d).

The finding of SMG5-dependent alterations of UPF1 phosphorylation levels prompted us to test whether, in cells lacking either SMG5, SMG7, or both, the general UPF1 phosphorylation status is changed. These alterations could reflect stalled UPF1-containing complexes in an arrested NMD processes. To this end, we expressed FLAG-tagged UPF1 in WT or SMG7 KO cells and detected its overall S/T phosphorylation level after immunoprecipitation (Extended Data Fig. 4e). The overexpressed UPF1 showed elevated phosphorylation in the SMG7 KO cells, which were not further increased when SMG5 was depleted (Extended Data Fig. 4e). We also assessed the SMG5-SMG7 dependent phosphorylation status of endogenous UPF1 using the phospho-S1127 specific antibody. Whereas the isolated KD of SMG5 had no effect on the phosphorylation status, similarly increased p-UPF1/UPF1 ratios were detected in SMG7 KO cells with or without SMG5 KD (Fig. 3h). Together, this indicates that the severe NMD defects observed in SMG5-SMG7 depleted cells compared to SMG7 KO cells do not coincide with further increased accumulation of hyper-phosphorylated UPF1.

Searching for other possible molecular explanations, we considered previous reports that SMG5 and SMG7 stabilize p-UPF1 binding to target mRNAs[61] and ATPase-deficient UPF1 mutants accumulate in a hyper-phosphorylated, SMG5-SMG7 bound state[48]. Inspired by these results, we investigated whether the ability of UPF1 to bind to and/or to recognize NMD targets is impaired in SMG5-SMG7 depleted cells. To this end, we employed UPF1 RNA immunoprecipitation (RIP) assays to study NMD target discrimination by UPF1[48]. Binding of UPF1 to two NMD targets, which displayed increased mRNA levels upon NMD inhibition (Supplementary Data 1), as well as to two non-NMD targets remained unchanged in control, SMG5 KD, SMG7 KO, or SMG7 KO plus SMG5 KD conditions (Extended Data Fig. 4f–i). These results suggest that UPF1 can identify and bind to NMD-targeted transcripts, although their degradation cannot be executed anymore.

**NMD is severely inhibited transcriptome-wide upon loss of SMG5-SMG7.** The strongly impaired NMD in cells depleted of SMG5 and SMG7 encouraged us to sequence mRNA from SMG7 KO cells (clones 2 and 34) with an additional SMG5 or SMG6 KD (Extended Data Fig. 5a). As expected for severe NMD inhibition, the combined depletion of SMG6 and SMG7 resulted in massive changes of gene expression and isoform usage, which were qualitatively and quantitatively comparable to published SMG6-SMG7 double KD in HeLa cells[14] (Fig. 4a–c and Extended Data Fig. 5b–d). Whereas SMG5 KD in control cells had very mild effects on the transcriptome, downregulation of SMG5 in SMG7

KO cells exhibited equal or even more pronounced changes in gene expression and isoform usage compared to the SMG6-SMG7-depleted condition (Fig. 4d–f and Extended Data Fig. 5e–g). As a representative example, the alternative splicing pattern of SRSF2 displayed a complete switch from the normal to NMD-sensitive isoforms when SMG5 or SMG6 were depleted in SMG7 KO cells (Fig. 4g). Interestingly, the highest overlap of upregulated NMD-sensitive isoforms was found between both SMG7 KO cell lines with SMG5 or SMG6 KD, suggesting that these four conditions predominantly target the same transcripts (Extended Data Fig. 5h).

Intrigued by the remarkable changes in gene expression and isoform usage when NMD was effectively inhibited, we wanted to re-examine the statement that up to 10% of genes are affected by NMD. To this end, we calculated how many genes showed single or combined differential gene expression (DGE), differential transcript usage (DTU), or alternative splicing (AS) events when the SMG7 KO was combined with the KD of SMG5 or SMG6 (Fig. 4h). With this approach, we find that about 40% of the expressed genome in the Flp-In-T-REx-293 cells is directly or indirectly altered by strong NMD inhibition. With more stringent cutoffs for the analyses, still, around 20% of all expressed genes were affected (Extended Data Fig. 5i), mostly those with medium to high expression levels (Extended Data Fig. 5j). Collectively, the RNA-Seq analysis confirmed that SMG5 KD, as well as SMG6 KD, have similar effects on NMD in SMG7 KO cells. It also provided global evidence that the loss of the SMG5-SMG7 heterodimer effectively inactivates NMD and leads to massive changes in the expressed transcriptome. In combination with the functional studies (Figs. 2–3), these transcriptome-wide observations profoundly question the independence of the SMG5-SMG7 and SMG6 decay pathways and suggest a functional connection during NMD execution.

**Loss of SMG5 and SMG7 prohibits endonucleolytic cleavage of NMD substrates.** The observed NMD inhibition upon the co-depletion of SMG5 and SMG7 suggests that SMG6 is equally inactivated, although it is widely assumed that SMG6 acts redundantly to and independently of SMG5-SMG7[14,64]. This unexpected result raised the question, whether SMG6 requires the presence of SMG5-SMG7 for endonucleolytic cleavage of its target mRNAs during NMD. We monitored the activity of SMG6 by northern blotting, which allows us to detect decay intermediates resulting from endonucleolytic or 5′–3′ exonucleolytic degradation. To this end, stably integrated triosephosphate isomerase (TPI) mRNA reporters were expressed as WT or NMD-inducing PTC160 variant in control or SMG7 KO cells with different combinations of siRNA-mediated knockdowns (Fig. 5a). The reporter mRNA also contained XRN1-resistant sequences (xrRNAs) in the 3′ UTR that produce meta-stable intermediates of 5′–3′ decay (called xrFrag) and thereby provided information about the extent and directionality of mRNA degradation[72,80]. Upon depletion of the major cytoplasmic 5′–3′ exonuclease XRN1, SMG6-generated endonucleolytic cleavage products (designated 3′ fragments) of the PTC-containing reporter mRNA are detected as an additional band (Extended Data Fig. 6a; lane 6). Of note, the isolated SMG7 KO or SMG6 KD resulted in a slight accumulation of the full-length reporter (Extended Data Fig. 6a; lanes 8 and 10), indicating partial NMD inhibition consistent with the literature and our earlier observations. However, we did not observe increased 3′ fragment levels upon loss of SMG7, which indicates that the SMG6 activity is not compensatory in the absence of SMG7 (Fig. 5b and Extended Data Fig. 6a, b; lane 14). While KD of SMG5 in control cells had no inhibitory effect on endonucleolytic cleavage, depletion of SMG5 in SMG7 KO cells

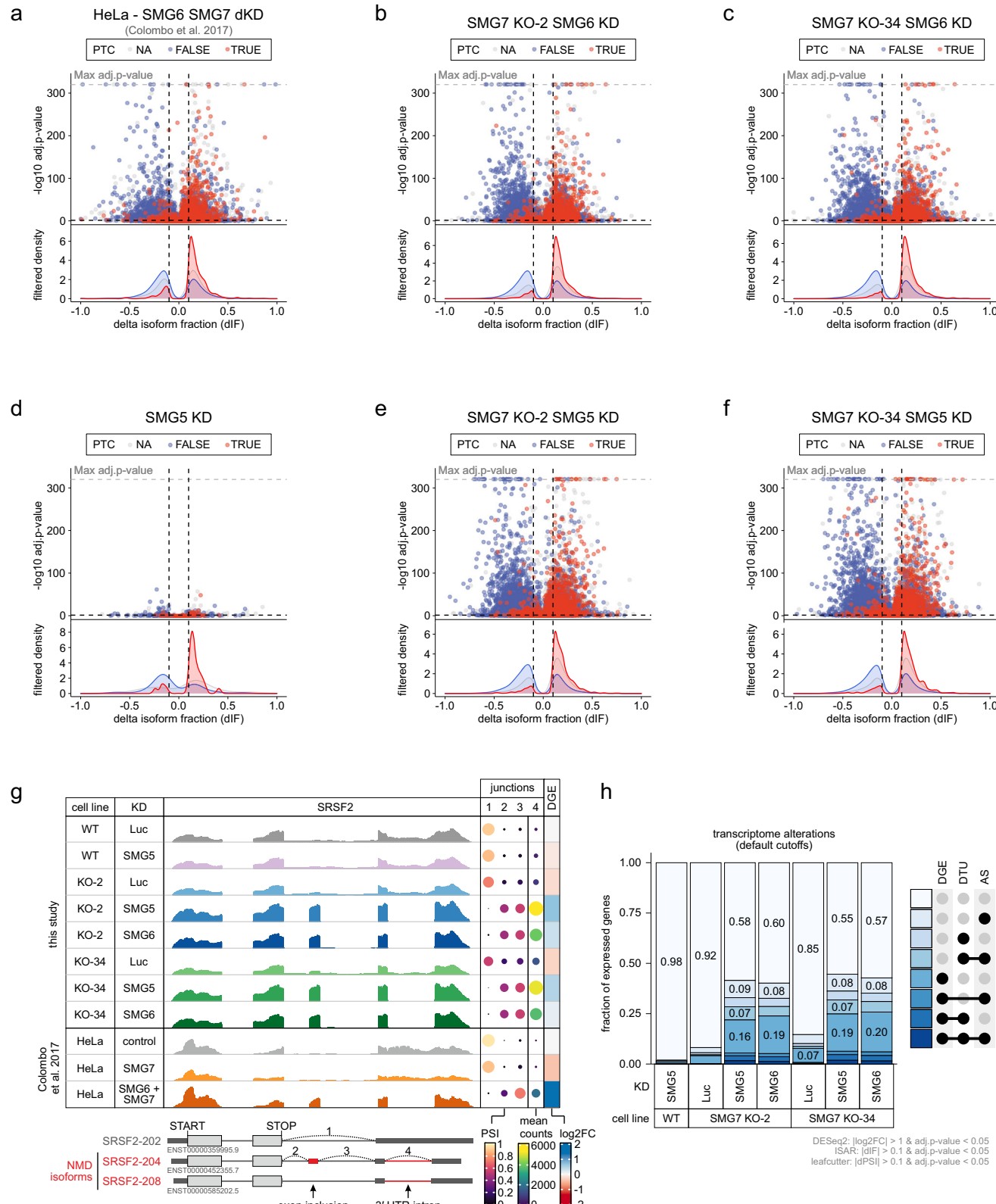

completely abolished the generation of 3′ fragments (Fig. 5b and Extended Data Fig. 6b; lanes 4 and 8 vs. 12 and 16). Furthermore, the accumulation of the PTC160 reporter mRNA to WT levels and the decrease of the xrFrag band confirmed that NMD is thoroughly inactivated in SMG5-SMG7-depleted conditions. The dramatic effect on the endonucleolytic cleavage activity

was further confirmed by investigating the naturally occurring meta-stable cleavage product of NOP56 (Fig. 5c)[81]. We used the NOP56 cleavage product as an indicator for SMG6 activity since the generation of this meta-stable RNA fragment was shown to be strongly SMG6-dependent[81]. In control cells, the cleavage product was predominantly present but became undetectable in

**Fig. 4 Downregulation of SMG5 in SMG7 knockout cells efficiently inactivates NMD. a–f** Volcano plots showing the differential transcript usage (via IsoformSwitchAnalyzeR) in various RNA-Seq data. Isoforms containing GENCODE (release 33) annotated PTC (red, TRUE), regular stop codons (blue, FALSE), or having no annotated open reading frame (gray, NA) are indicated. The change in isoform fraction (dIF) is plotted against the −log10 adjusted *p*-value (adj.*p*-value). Density plots show the distribution of filtered isoforms in respect to the dIF, cutoffs were |dIF|> 0.1 and adj.*p*-value <0.05. *P*-values were calculated by IsoformSwitchAnalyzeR using a DEXSeq-based test and corrected for multiple testing using the Benjamini-Hochberg method. **g** Read coverage of SRSF2 from SMG7 KO and published SMG7 KD (GEO: GSE86148) RNA-Seq data is shown as Integrative Genomics Viewer (IGV) snapshots. The canonical and NMD-sensitive isoforms are schematically indicated below. Percent spliced in (PSI; from LeafCutter analysis) and mean counts from 4 indicative splice junctions are shown. Differential gene expression (from DESeq2) is depicted as log2 fold change (log2FC) in the last column. **h** Fraction of expressed genes (genes with non-zero counts in DESeq2) were calculated which exhibit individual or combinations of differential gene expression (DGE), differential transcript usage (DTU), and/or alternative splicing (AS) events in the indicated conditions using the respective computational analysis (default cutoffs are indicated). AS and DTU events were collapsed on the gene level. For DGE, *p*-values were calculated by DESeq2 using a two-sided Wald test and corrected for multiple testing using the Benjamini-Hochberg method. For DTU, *p*-values were calculated by IsoformSwitchAnalyzeR using a DEXSeq-based test and corrected for multiple testing using the Benjamini-Hochberg method. For AS, *p*-values were calculated by LeafCutter using an asymptotic Chi-squared distribution and corrected for multiple testing using the Benjamini-Hochberg method.

the SMG5-SMG7-depleted condition (Fig. 5d). Taken together, these results underline the previous observation that the SMG5-SMG7 heterodimer is required for general NMD activity and, surprisingly, also for SMG6 activity.

Finally, we were intrigued by the fact that both SMG5 and SMG7 wild-type proteins can individually rescue the SMG7 KO phenotype. Specifically, we wondered if the main NMD-supporting function of both factors could be to enable and/or sustain SMG6-mediated endonucleolytic cleavage of the target mRNA. All SMG5 and SMG7 rescue proteins that restored full NMD activity also resulted in a normal NOP56 cleavage pattern, indicating that SMG6 was reactivated in these cells (Fig. 5d). Based on these results, we postulate that SMG5 and SMG7 maintain NMD activity by permitting the activation of SMG6.

**SMG6 can interact with UPF1 in SMG5-SMG7-depleted cells.** Given the inability of SMG6 to cleave target mRNAs in SMG5-SMG7 depleted cells, we next investigated the interaction between SMG6 and UPF1, which we considered central in this context. Due to the low abundance of SMG6 compared to UPF1 (more than 100-fold lower; Extended Data Fig. 7a) and the presumably transient mode of their interaction, the investigation of UPF1-SMG6 binding in the cellular context has been notoriously difficult to detect by standard co-IP experiment[82]. To detect the transient interaction between SMG6 and UPF1, we used the TurboID-catalyzed proximity labeling technology[83,84] to biotinylate UPF1 binding partners (Fig. 6a). We established stable cell lines that inducibly expressed FLAG-TurboID (reference construct) or FLAG-TurboID-UPF1 and confirmed the biotinylation activity of these constructs (Extended Data Fig. 7b). We used 15 min biotin labeling for both TurboID constructs in WT and SMG7 KO cells upon control or SMG5 KD (Fig. 6b). FLAG-TurboID-UPF1 displayed prominent self-biotinylation but also efficiently biotinylated the known binding partner UPF3B in contrast to the TurboID control (Fig. 6b). Nevertheless, we were not able to obtain quantitative data regarding the interaction of SMG6 with UPF1 by western blotting. To overcome this problem, we turned to the label-free mass spectrometric analysis of the streptavidin-enriched proteins (Supplementary Data 5). We detected known UPF1 interacting NMD factors (e.g., UPF3B), but also other binding partners (e.g., Staufen proteins; STAU1 and STAU2)[85,86], to be enriched in FLAG-TurboID-UPF1-expressing WT cells, indicating that we indeed captured true UPF1 interaction partners (Extended Data Fig. 7d and Supplementary Data 5). Interestingly, we found in SMG7 KO cells a higher enrichment of almost all NMD factors, including the NMD proteins SMG1, SMG6, SMG8, and SMG9, which were not enriched in WT cells. The enrichment was statistically even more pronounced with an additional SMG5 KD (Fig. 6c–e and

Extended Data Fig. 7c–f). These findings indicate that UPF1 accumulates together with NMD factors (including SMG6) in functionally arrested complexes in the absence of SMG5 and SMG7. Since SMG6 was apparently catalytically inactive in SMG5-SMG7 depleted cells (Fig. 5), this observed UPF1-SMG6 interaction is likely unproductive and does not represent a functional NMD complex.

## Discussion

The correct execution of NMD not only prevents the production of aberrant gene products but also shapes the transcriptome on a global scale[43]. NMD is generally perceived as a robust, but highly dynamic process that integrates different inputs, including mRNP composition and translational status, in order to efficiently identify and remove transcripts that appear to be faulty[42]. The common perception of NMD is that multiple RNA degradation pathways can be employed after the identification of target transcripts, which are all centered around the key factor UPF1 and provide reliable elimination of the mRNA. The previously identified two major decay paths during NMD utilize the UPF1-recruited SMG5-SMG7 and SMG6. Although evidence pointed to the independence of these branches in the past, we show here that SMG6 cannot endonucleolytically cleave NMD substrates in cells lacking the SMG5-SMG7 heterodimer (Fig. 5 and Extended Data Fig. 6), resulting in extensive NMD inactivation. Therefore, a functional dependency between the two pathways exists.

The reason why this dependency has not been detected so far has probably technical reasons. Virtually all previous experiments that addressed the interplay between SMG5, SMG6 and SMG7 utilized individual or combined gene silencing of NMD factors depending on siRNA-mediated or shRNA-mediated knockdown. As reported before, the downregulation of SMG5 and/or SMG7 by knockdowns only slightly impairs NMD[35,62]. We show here that complete and sustained depletion of SMG7 is needed to detect a considerable NMD defect (Fig. 1 and Extended Data Fig. 2). Furthermore, the downregulation of SMG5 substantially affects NMD activity only in the SMG7 KO conditions (Figs. 2, 4 and Extended Data Fig. 5). Therefore, we propose that a conventional downregulation of the SMG5-SMG7 heterodimer is not sufficient to abolish its function. It was reported before that SMG5 and SMG7 form stable and long-lived complexes[35] and residual heterodimers could potentially outlive the experimental timeframe of knockdown experiments. Remarkably, strongly reduced levels of the SMG5-SMG7 heterodimer after a knockdown are still able to support NMD, although both proteins are about two orders of magnitude less abundant than UPF1 (Extended Data Fig. 7a)[87–89]. This observation indicates that the basal levels of SMG5 and SMG7 provide enough buffer capacity to tolerate the partial loss of individual NMD factors or to cope

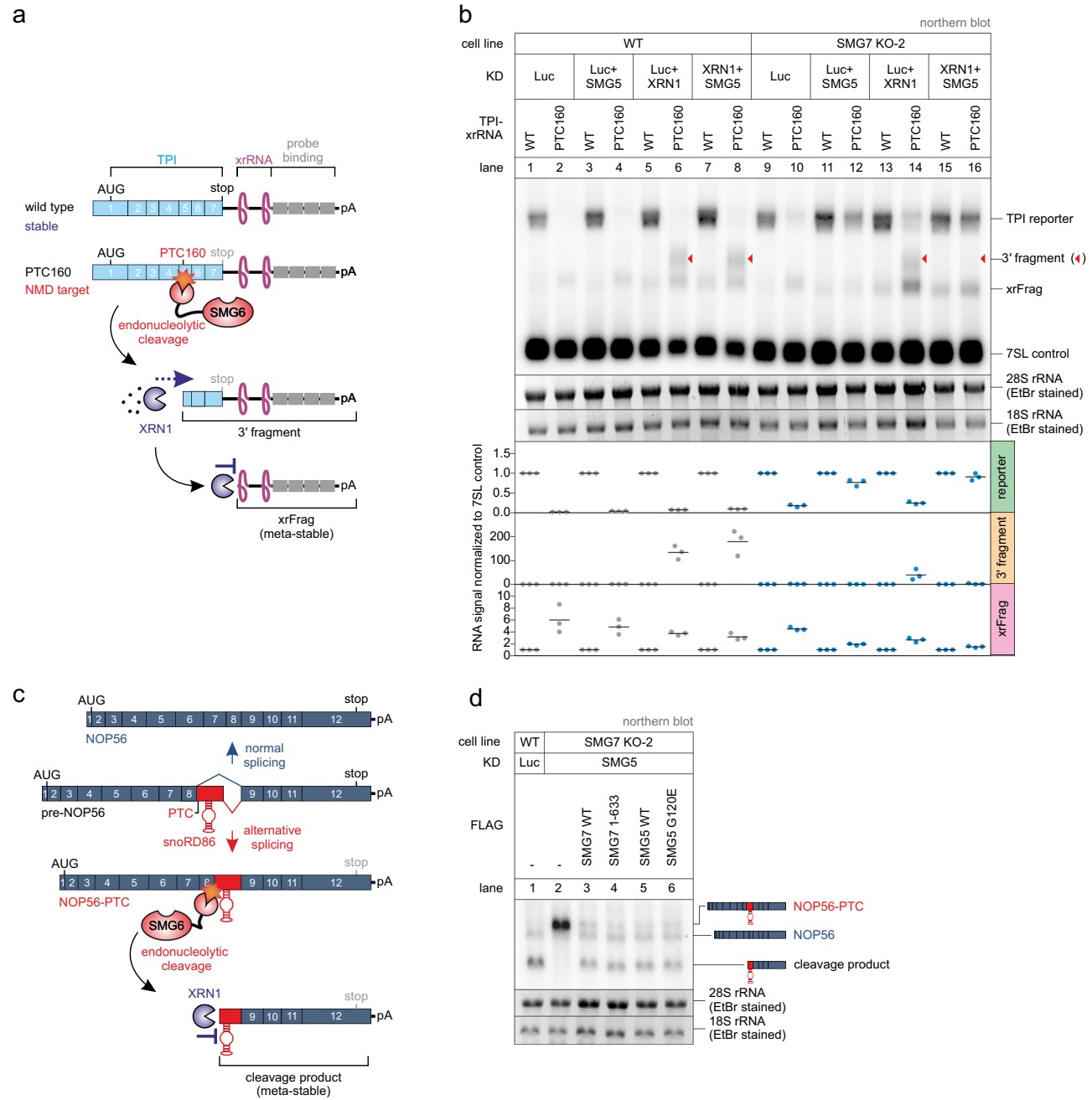

**Fig. 5 SMG6 endonucleolytic cleavage is inactivated in SMG5-SMG7 depleted cells. a** Schematic overview of the triosephosphate isomerase (TPI) reporter constructs and their functional elements. The PTC160-containing reporter is subjected to SMG6-mediated endonucleolytic cleavage during NMD. The resulting decay intermediate (3′ fragment) is rapidly degraded by XRN1, until the presence of an xrRNA blocks the processivity of XRN1, ultimately resulting in the accumulation of meta-stable xrFrag molecules. **b** Northern blot analysis of TPI reporter, 3′ fragments (indicated with red triangles), xrFrag, and 7SL endogenous control. Ethidium bromide-stained 28S and 18S rRNAs are shown as additional controls. Quantification results (normalized to 7SL control) are shown as data points and mean ($n = 3$ biologically independent samples). **c** Schematic depiction of NOP56 maturation and the consequences of alternative splicing for inducing NMD. Upon inclusion of the snoRD86 sequence in the mature NOP56 mRNA by alternative splice site usage, a PTC is introduced resulting in endonucleolytic cleavage by SMG6. XRN1-mediated degradation of the decay intermediate is hindered by snoRD86, resulting in the accumulation of this meta-stable cleavage product. **d** Northern blot analysis of endogenous NOP56 ($n = 3$ biologically independent samples). Different transcript isoforms and fragments are indicated.

with increasing amounts of NMD targets, e.g., resulting from reduced transcriptomic fidelity. In line with this idea, previous attempts to "overload" the NMD machinery by transiently overexpressing large quantities of NMD substrates did not result in reduced NMD activity[90].

The remarkable capacity of the NMD process is also reflected in the amount of differentially regulated transcripts that accumulate in cells with inactive NMD. Earlier studies estimated that about 5–10% of all human genes are directly or indirectly influenced by NMD[9,12–15]. If we consider gene-specific and isoform-specific effects (differential gene expression, isoform switches, alternative splicing), we find that between 20 and 40% of the expressed genes are affected by NMD. These values are considerably higher than previous estimates, which can be

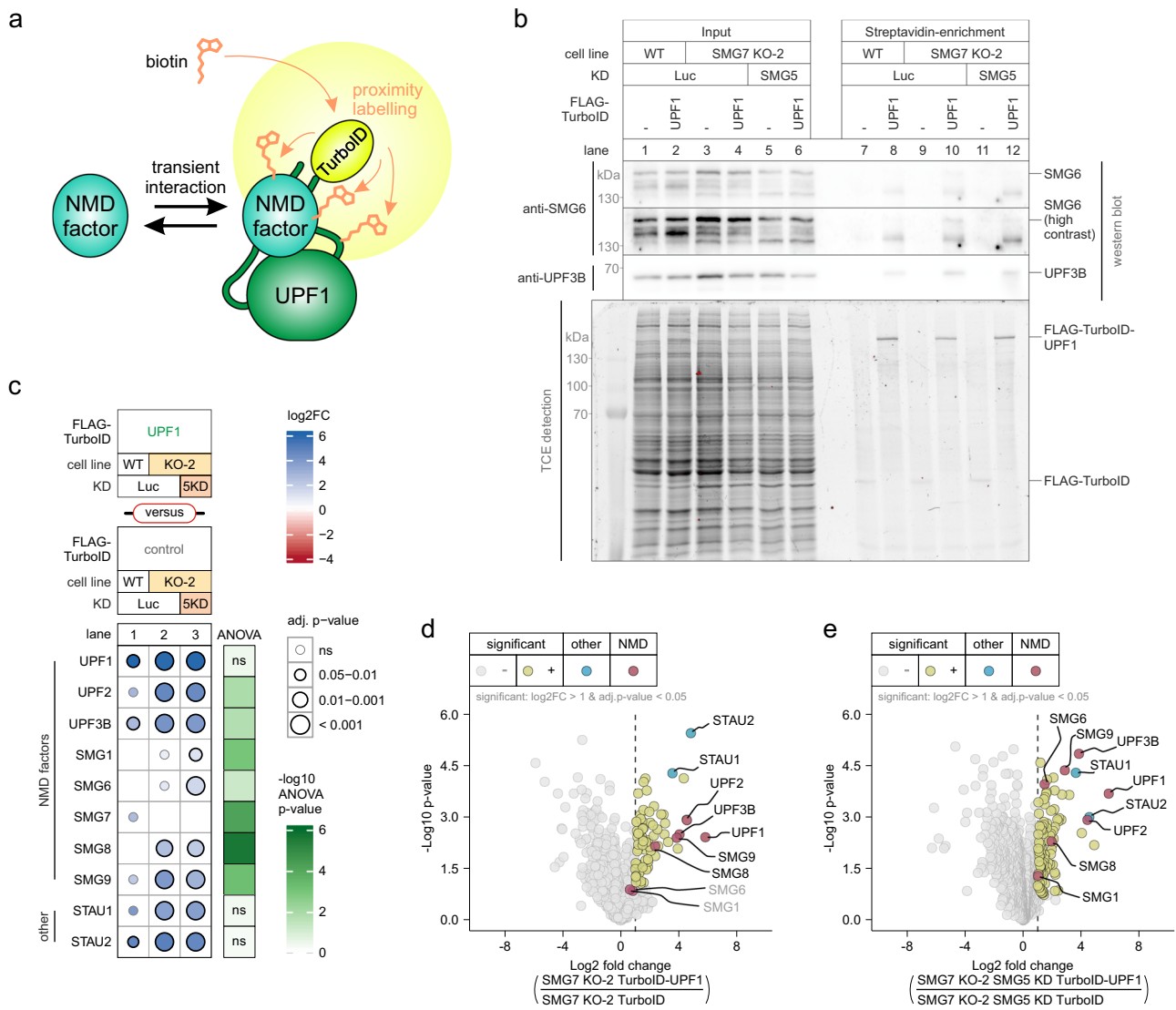

**Fig. 6 UPF1 accumulates with NMD factors in SMG5-SMG7 depleted cells. a** Overview of the TurboID-mediated proximity labeling of UPF1 binding partners. Transient UPF1 interactors are marked with biotin via TurboID catalysis. Biotinylated proteins are subsequently enriched with streptavidin beads. **b** Western blot and TCE (2,2,2-trichloroethanol)-mediated detection of input or streptavidin-enriched protein samples after proximity labeling in the indicated conditions ($n = 3$ biologically independent samples). **c** Heatmap of mass spectrometry-based analysis of streptavidin-enriched biotinylated NMD and selected other proteins in the respective comparison of conditions ($n = 3$ biologically independent samples). Colored points indicate the log2 fold change (log2FC) and point size corresponds to the adjusted $p$-value (adj. $p$-value; from two-sided Welch's $t$-test). Multiple testing between FLAG-TurboID-UPF1 in WT, SMG7 KO, and SMG7 KO + SMG5 KD was performed by ANOVA. **d, e** Volcano plots of mass spectrometry-based analysis of streptavidin-enriched biotinylated proteins in the respective comparison of conditions ($n = 3$ biologically independent samples). **d** FLAG-TurboID-UPF1 against FLAG-TurboID control in SMG7 KO cells, (**e**) FLAG-TurboID-UPF1 against FLAG-TurboID control in SMG7 KO + SMG5 KD cells. The yellow color labeling indicates targets that are significant in the respective comparisons after two-sided Welch's $t$-testing (log2 fold change (log2FC) >1 or |log2FC| >1; and adj. $p$-value <0.05). Points labeled in blue indicate other proteins of interest; points labeled in red indicate NMD factors. Highlighted proteins that were not significant in the respective comparisons are labeled with gray text.

partially explained by using state-of-the-art RNA-sequencing methods and recent bioinformatic algorithms, allowing a more thorough analysis of the transcriptomic alterations. Furthermore, we believe that the SMG7 KO cells in combination with SMG5 or SMG6 KD result in a more efficient NMD inhibition, which could not be achieved with previous attempts based on RNA interference alone. Admittedly, not all of the detected changes in NMD-incompetent cells are direct effects of NMD inhibition, since the misregulation of targets such as the splicing factor SRSF2 will undoubtedly cause secondary effects on the transcriptome. However, the large number of NMD-regulated genes can explain why NMD is essential for cell survival, proliferation

and differentiation. It is difficult to imagine that important and fundamental biological processes can function normally when about one-third of all expressed genes are affected. Given this large amount of potential cellular NMD substrates, it will be important in the future to identify and characterize which mRNA isoforms are authentic NMD-regulated transcripts. This will also help to better understand the process of NMD and establish further rules for NMD activating features[91].

Our detailed characterization of SMG5 and SMG7 individual functions revealed interesting and unexpected insights into the function of these two proteins. Most impressively, the two proteins seem to exhibit a certain redundancy, because SMG5 or

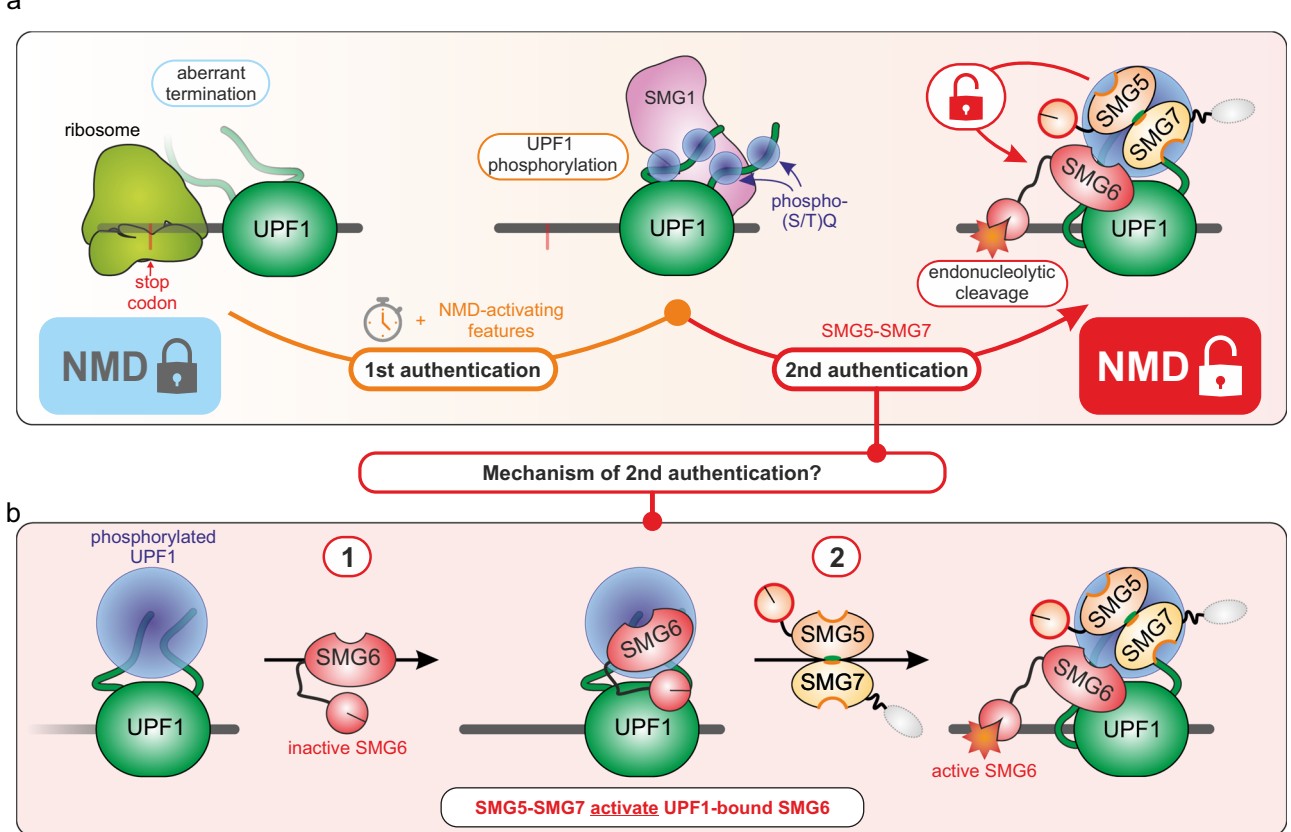

**Fig. 7 Two-factor authentication model of NMD. a** Overview of postulated two-factor authentication model. To grant access to the degradative activities of NMD, at least two consecutive security checks need to be passed. Aberrant translation termination in combination with an enhanced residence time of UPF1 on the mRNA and NMD-activating features (e.g., the presence of a downstream EJC) lead to increasing UPF1 phosphorylation by SMG1 (1st authentication step). Sufficiently phosphorylated UPF1 (p-UPF1) leads to the recruitment of SMG5-SMG7 and consequently enables the endonucleolytic activity of SMG6 (2nd authentication step). Domains and functions of SMG5 and SMG7 required to support NMD are indicated (green = low importance, orange = medium, red = high). **b** Hypothesis for the potential mechanism of the second authentication step. According to this hypothesis, SMG6 can be bound to p-UPF1 before SMG5-SMG7. The role of SMG5-SMG7 in NMD would be the activation of p-UPF1-bound SMG6.

SMG7, respectively, rescue the combined SMG5-SMG7 depletion. The simplest explanation would be that both proteins could form homodimers in the absence of the heterodimerization partner. Although not excluded, this possibility does not seem very likely based on our own preparatory data. Our functional character-ization established that the main function of SMG5-SMG7 cannot simply be the recruitment of deadenylation-promoting factors as was previously proposed[64]. Also, we could not confirm the direct recruitment of PP2A phosphatases via the C-terminal PIN domain of SMG5. Therefore, the overlapping role of SMG5 and SMG7 during NMD activation probably represents their most important molecular function. While a single factor (e.g., SMG7) would presumably be able to execute the same function when expressed at higher levels, such a system could potentially also lead to more unwanted NMD activation on barely phosphory-lated UPF1. Accordingly, SMG5-SMG7 may balance robust degradation of NMD substrates on the one hand with tight control of NMD activity on the other.

On a conceptional level, NMD identifies and degrades tran-scripts that fail to pass quality control standards. To this end, the NMD pathway makes use of potent cytoplasmic RNA degrada-tion tools, such as the endonuclease SMG6. However, the access to and application of this tool must be very tightly controlled to minimize spurious degradation of normal transcripts. This con-trol is especially important since NMD probably monitors every translation termination event and uncontrolled SMG6-mediated

mRNA cleavage would be catastrophic for the cell. Based on our results, we propose an improved model for the activation of NMD and attempt to integrate and reconcile earlier observations. In our opinion, this can best be accomplished with a two-factor authentication model (Fig. 7a), a security procedure in which two distinct credentials are required for proper identification. By analogy, UPF1 represents in our model a surveillance factor, which has to successfully pass at least two different consecutive authentication steps on true NMD targets to gain access to SMG6-mediated activity.

We propose that UPF1 is routinely assigned to control the quality of all translated mRNAs. It has become increasingly clear that the mere binding of UPF1 to an mRNA does not trigger its degradation. However, UPF1 must be removed from the tran-script in a timely manner to prevent the formation of NMD-activating complexes. Towards this goal, translating ribosomes remove most of the RNA-bound UPF1, and only those in the 3' UTR remain attached to the transcript[44–47]. Furthermore, effi-cient and proper translation termination results in the dissocia-tion of UPF1 from the transcript[48].

When translation terminates inefficiently or at aberrant posi-tions (e.g., upstream of an EJC), this NMD-promoting environ-ment increases the residence time of UPF1 on the mRNA. At this point, the first authentication step is installed, which is the pro-gressive SMG1-mediated phosphorylation of UPF1 (Fig. 7a). It has been shown that SMG1 associates with and phosphorylates

UPF1 preferentially in the presence of UPF2, UPF3, and the EJC[50,52,57,58,92]. This ensures that UPF1 is only phosphorylated when bound to positions on the mRNA, where an NMD-activating arrangement of mRNP features is present. Effectively, UPF1 remains attached to the (so far only putative) NMD-transcript and persisting NMD-activating features will cumulatively lead to hyper-phosphorylated UPF1 (Fig. 7a).

How is the increasing phosphorylation ultimately converted to an appropriate mRNA degradation response? We propose, based on the existing literature and our data, that a second authentication procedure must be passed to gain access to and/or activate SMG6 (Fig. 7a). Accordingly, we envision an interaction of SMG5-SMG7 (individually or as a heterodimer) with phosphorylated UPF1 as an essential step to permit efficient SMG6-mediated endonucleolytic cleavage of the target mRNA. The evidence supporting this general interpretation is foremost the robust NMD inhibition in SMG5-SMG7-depleted cells, with no detectable SMG6-activity (Figs. 2, 4, 5). Furthermore, the transcriptomic changes are highly similar between the loss of the SMG5-SMG7 heterodimer and the combined SMG7 KO with SMG6 KD, indicating a similar functional outcome. Further support for this model comes from the observation that hyper-phosphorylation alone is not sufficient to induce NMD, as ATPase-deficient UPF1 mutants are phosphorylated but do not support NMD[42,50,59]. Likewise, we also find that globally increased UPF1 phosphorylation levels do not positively correlate with NMD activity, as seen in SMG7 KO cells or upon expression of the SMG5 1-853 mutant (Fig. 3 and Extended Data Fig. 4).

This raises the question about the molecular mechanism behind the second authentication step and why it depends on SMG5-SMG7. We initially proposed three hypothetical models for this second authentication step that could explain the function of SMG5-SMG7 in NMD. These three models differed foremost in the way how the access of SMG6 to UPF1 is controlled. Although SMG6 contains a 14-3-3-like domain and was proposed to interact directly with UPF1 via phosphorylation-dependent binding[60], a phosphorylation-independent interaction between SMG6 and UPF1 was reported afterwards[63,77,93]. If this phosphorylation-independent SMG6-UPF1 interaction occurred constantly and was sufficient to activate SMG6-mediated endonucleolytic cleavage, uncontrolled NMD would be observed with no target discrimination. Since this is not the case, we rather envision that the binding of SMG5-SMG7 to p-UPF1 could be required to establish the interaction of SMG6 with UPF1 on NMD targets. This would fit with the assumption that UPF1 needs to be sufficiently phosphorylated and bound by SMG5-SMG7 in order to allow SMG6 activity. However, strongly contradicting this hypothesis, we do not observe less, but rather more SMG6 interaction with UPF1 in SMG5-SMG7-depleted cells.

According to our second hypothesis, catalytically active SMG6 could directly interact with sufficiently phosphorylated UPF1 independently of SMG5-SMG7 and initiate the degradation of the target mRNA via endonucleolysis. However, the interaction of SMG5-SMG7 with p-UPF1 would be required to resolve this NMD-complex and to liberate SMG6 from UPF1. The mechanism could be the displacement of SMG6 from phosphorylated UPF1 residues by SMG5-SMG7. The absence of SMG5-SMG7 or their inability to interact with p-UPF1 would lead to dead-end UPF1-SMG6 complexes that are stuck on cleaved mRNA fragments and cannot engage other NMD targets. Especially considering the low abundance of SMG6 (Extended Data Fig. 7a), this would effectively trap SMG6, shut down NMD activity, and lead to practically undetectable endonucleolytic cleavage, as observed in our northern blot experiments (Fig. 5).

The last and in our opinion most favorable model is very similar to the second, except that SMG6 is at first inactive when bound to p-UPF1 and the subsequent interaction of SMG5-SMG7 with p-UPF1 is required to activate SMG6 (Fig. 7b). According to this hypothesis, increased interaction of SMG6 with p-UPF1 should occur in SMG5-SMG7-depleted cells, as we have observed in the UPF1 proximity labeling experiments (Fig. 6). The way in which SMG5-SMG7 could activate UPF1-bound SMG6 remains to be investigated in more detail. Possible mechanisms include a conformational change of UPF1 and/or a switch in the SMG6-UPF1 binding mode upon binding of SMG5-SMG7. In any case, according to this hypothesis, SMG6 would remain inactive until SMG5-SMG7 sensed that UPF1 is sufficiently phosphorylated. This NMD-routine would ensure that UPF1 remains locked on transcripts that require further inspection until a decision has been made to either release the surveillance factor UPF1 or to degrade the mRNA. Therefore, in the absence of SMG5-SMG7, UPF1 molecules that engaged NMD authentication would accumulate in unproductive complexes, being unable to dissociate from the mRNA or to initiate target degradation. Interestingly, we observe this effect in our TurboID experiment (Fig. 6). When SMG5 and SMG7 are absent, several NMD factors involved in the first authentication (e.g., SMG1, SMG8, and SMG9) are biotinylated more by TurboID-UPF1, indicating a stronger interaction with UPF1 or a longer residence time in the proximity of UPF1. Whether these enriched NMD factors represent one specific or more diverse NMD complexes that accumulate on target substrates cannot be conclusively resolved. Since SMG6 seems to also interact more with UPF1, we speculate that the loss of SMG5-SMG7 could allow more SMG6 to unproductively bind phosphorylated UPF1, which phospho-sites are no longer occupied by SMG5-SMG7. Although further research would be needed to formulate a definitive model, we believe that combined aspects from the three above discussed models may come close to explaining the essential function of SMG5-SMG7 during NMD.

In conclusion, we present here a revised model for the activation and execution of NMD. This model is centered around UPF1 and involves progressive SMG1-mediated phosphorylation as first, and SMG5-SMG7-mediated activation/recycling of SMG6 as the second authentication step to identify and degrade NMD targets in a complex transcriptome. The proposed roles of NMD factors in our model create ample opportunities to investigate their function and interplay and allows the field to move away from earlier models, which were based on parallel or redundant degradation pathways during NMD.

## Methods

**Cell lines**. Flp-In-T-REx-293 (human, female, embryonic kidney, epithelial; Thermo Fisher Scientific, RRID:CVCL_U427) cells were cultured in high glucose, GlutaMAX DMEM (Gibco) supplemented with 9% fetal bovine serum (Gibco) and 1 × Penicillin Streptomycin (Gibco). The cells were cultivated at 37 °C and 5% $CO_2$ in a humidified incubator. The generation of knockout and stable cell lines is described below and all cell lines are summarized in Supplementary Data 6.

**Generation of knockout cells using CRISPR-Cas9**. The knockouts were performed using the Alt-R CRISPR-Cas9 system (Integrated DNA Technologies) and reverse transfection of a Cas9:guideRNA ribonucleoprotein complex using Lipofectamine RNAiMAX (Thermo Fisher Scientific) according to the manufacturer's protocol. The crRNA sequence (Design ID: Hs.Cas9.SMG7.1.AD; Integrated DNA Technologies) to target SMG7 was /AltR1/rGrArArArArArUrGrCrUrArGrUrUrAr CrCrGrArUrUrGrUrUrUrUrArArGrArGrCrUrArUrGrCrU/AltR2/. Reverse transfection was performed on $1.5 \times 10^5$ cells per crRNA in 12-well plates. 48 h after transfection the cells were trypsinized, counted, and seeded at a mean density of a single cell per well in 96-well plates. Cell colonies originating from a single clone were then screened via western blot and genome editing of SMG7 was analyzed on the genomic level via DNA extraction and Sanger sequencing (Eurofins Genomics). Alterations on the transcript level were analyzed via RNA extraction (see below) followed by reverse transcription and Sanger sequencing.

**DNA and RNA extraction**. One day prior to DNA extraction, $2.5 \times 10^5$ cells were seeded in a 6-well plate. To extract DNA, QuickExtract DNA Extraction Solution

(Lucigen) was used following the manufacturer's instructions. RNA was isolated using peqGOLD TriFast (VWR Peqlab) or RNA-Solv (Omega Bio-Tek) following the manufacturer's instructions. Following changes were made: Instead of 200 µl chloroform, 150 µl 1-Bromo-3-chloropropane (Molecular Research Center, Inc.) was used. RNA was resuspended in 20 µl RNase-free water.

**Immunoblot analysis**. SDS-polyacrylamide gel electrophoresis and immunoblot analysis were performed on protein samples harvested with RIPA buffer (50 mM Tris/HCl pH 8.0, 0.1% SDS, 150 mM NaCl, 1% IGEPAL CA 630, 0.5% deoxycholate) or samples eluted from Anti-FLAG M2 magnetic beads (Sigma-Aldrich). For protein quantification, the Pierce Detergent Compatible Bradford Assay Reagent (Thermo Fisher Scientific) was used. All antibodies were used at the indicated dilutions in 50 mM Tris [pH 7.2], 150 mM NaCl with 0.2% Tween-20, and 5% skim milk powder. Amersham ECL Prime or Select Western Blotting Detection Reagent (GE Healthcare) in combination with the Fusion FX-6 Edge system (Vilber Lourmat) was used for visualization. All antibodies used in this study are listed in Supplementary Data 6. Protein bands detected with the Fusion FX-6 Edge system (Vilber Lourmat) using the Evolution-Capt Edge software (version 18.05) were quantified in a semi-automated manner using the Image-Quant TL 1D software (version 8.1) with a rolling-ball background correction. The control condition was set to unity, quantification results are shown as data points and mean.

**Growth assay**. To measure the growth and mortality rate of cells, CytoTox-Glo Cytotoxicity Assay (Promega) was performed. 10,000 cells/well were seeded in a 96-well plate and the assay was performed after 0–4 days using luminometer Centro XS3 LB 960 (Berthold Technologies) and the MikroWin software (version 5.14) following the manufacturer's instructions.

**Stable cell lines and plasmids**. The point and deletion mutants of SMG7 were PCR amplified using Q5 polymerase (New England Biolabs) and inserted with an N-terminal FLAG-tag via NheI and NotI (both New England Biolabs) restriction sites into PB-CuO-MCS-IRES-GFP-EF1α-CymR-Puro (System Biosciences). Accordingly, N-terminally FLAG-tagged GST or UPF1, as well as FLAG-TurboID or FLAG-TurboID-UPF1 (generated with Integrated DNA Technologies gBlocks and/or PCR amplification) were cloned via NheI and NotI into PB-CuO-MCS-BGH-EF1-CymR-Puro, which was modified from the original vector by replacing the IRES-GFP cassette with a BGH polyA signal.

The point and deletion mutants of SMG5 were PCR amplified using Q5 polymerase and inserted with an N-terminal FLAG-tag via NheI and NotI restriction sites into the tetracycline-inducible pcDNA5/FRT/TO vector (Thermo Fisher Scientific). The mRNA reporter constructs TPI-WT and TPI-PTC160 in the pcDNA5/FRT/TO vector are available on Addgene (IDs 108377-108378).

The cells were stably transfected using the PiggyBac Transposon system (SMG7, UPF1, GST) or using the Flp-In T-REx system (SMG5, mRNA reporter). 2.5–3 × $10^5$ cells were seeded 24 h before transfection in 6-wells. For PiggyBac stable cells, 2 µg of PiggyBac construct was transfected together with 0.8 µg of the Super PiggyBac Transposase expressing vector and for Flp-In T-REx stable cells, 1-2 µg of pcDNA5 construct was transfected together with 1 µg of the Flp recombinase expressing plasmid pOG44, using the calcium phosphate method. Forty-eight hours after transfection, the cells were transferred into 10 cm dishes and selected with 2 µg ml$^{-1}$ puromycin (InvivoGen) for PiggyBac or 100 µg ml$^{-1}$ hygromycin (InvivoGen) for Flp-In T-REx. After 7–10 days, the colonies were pooled. Expression of the PiggyBac constructs was induced with 30 µg ml$^{-1}$ cumate, Flp-In T-REx constructs were induced with 1 µg ml$^{-1}$ doxycycline. All vectors used in this study are listed in Supplementary Data 6.

Mycoplasma contamination was tested by PCR amplification of mycoplasma-specific genomic DNA[94] or by using the Mycoplasmacheck service (Eurofins Genomics).

**Reverse transcription, end-point, and quantitative RT-PCR**. 1–4 µg of total RNA was reverse-transcribed in a 20 µl reaction volume with 10 µM VNN-(dT)$_{20}$ primer using the GoScript Reverse Transcriptase (Promega). 2% of cDNA was used as template in end-point PCRs using the GoTaq Green Master Mix (Promega) or MyTaq Red Mix (Bioline) and 0.2–0.6 µM final concentration of sense and antisense primer (see Supplementary Data 6 for sequences). After 30 PCR cycles, the samples were resolved by electrophoresis on ethidium bromide-stained, 1–2% agarose TBE gels and visualized by trans-UV illumination using the Gel Doc XR+ (Bio-Rad) and Image Lab software (version 5.1).

Bands detected in agarose gels from the indicated biological replicates of end-point PCRs were quantified using the Image Lab software (version 6.0.1). Results of the indicated band % per lane are shown as data points and mean. Sanger sequencing of individual bands was performed using the service of Eurofins Genomics.

Quantitative RT-PCR was performed with the GoTaq qPCR Master Mix (Promega) using 2% of cDNA in 10 µl reactions, 0.2–0.6 µM final concentration of sense and antisense primer (see Supplementary Data 6 for sequences), and the CFX96 Touch Real-Time PCR Detection System (Bio-Rad) with Bio-Rad CFX Manager software (version 3.0). The reactions for each biological replicate were

performed in duplicates or triplicates and the average Ct (threshold cycle) value was measured. For alternative splicing events, values for canonical isoforms were subtracted from values for NMD-sensitive isoforms to calculate the ΔCt. For differentially expressed targets, the values for the housekeeping genes C1orf43 were subtracted from the values for the target to calculate the ΔCt. The mean log2 fold changes were calculated from three biologically independent experiments. Log2 fold change results are shown as data points and mean.

**siRNA-mediated knockdowns**. Cells were seeded in 6-well plates at a density of 2–3 × $10^5$ cells per well and reverse transfected using 2.5 µl Lipofectamine RNAi-MAX and 60 pmol of the respective siRNA(s) according to the manufacturer's instructions. In preparation for UPF1 phosphorylation and RNA immunoprecipitation (RIP) assays, 3 × $10^6$ cells were reverse transfected in 10 cm dishes using 6.25 µl Lipofectamine RNAiMAX and 150–200 pmol siRNA. All siRNAs used in this study are listed in Supplementary Data 6.

**RNA-sequencing and computational analyses**. RNA-Seq experiments were carried out with Flp-In-T-REx-293 wild-type (WT) cells transfected with Luciferase or SMG5 siRNA and the SMG7 KO clones 2 and 34 transfected with either Luciferase, SMG5, or SMG6 siRNAs. Three biological replicates were analyzed for each sample. Total RNA was extracted using peqGOLD TriFast (VWR Peqlab) as described above.

The Lexogen SIRV Set1 Spike-In Control Mix (SKU: 025.03) that provides a set of external RNA controls was added to the total RNA to enable performance assessment. Mix E0 was added to samples with Luciferase siRNA, mix E1 was added to samples with SMG5 siRNA, and mix E2 samples with SMG6 siRNA. The Spike-Ins were used for quality control purposes, but not used for the final analysis of DGE, DTU, or AS.

The library preparation was performed with the TruSeq mRNA Stranded kit (Illumina). After poly-A selection (using poly-T oligo-attached magnetic beads), mRNA was purified and fragmented using divalent cations under elevated temperatures. The RNA fragments underwent reverse transcription using random primers. This is followed by second-strand cDNA synthesis with DNA Polymerase I and RNase H. After end repair and A-tailing, indexing adapters were ligated. The products were then purified and amplified to create the final cDNA libraries. After library validation and quantification (Agilent tape station), equimolar amounts of the library were pooled. The pool was quantified by using the Peqlab KAPA Library Quantification Kit and the Applied Biosystems 7900HT Sequence Detection System and sequenced on an Illumina NovaSeq6000 sequencing instrument and a PE100 protocol.

Reads were aligned against the human genome (version 38, GENCODE release 33 transcript annotations[95] supplemented with SIRVomeERCCome annotations from Lexogen; obtained from https://www.lexogen.com/sirvs/download/) using the STAR read aligner (version 2.7.3a)[96]. Transcript abundance estimates were computed with Salmon (version 1.3.0)[97] with a decoy-aware transcriptome. After the import of transcript abundances, differential gene expression analysis was performed with the DESeq2[98] R package (version 1.28.1) with the significance thresholds |log2FoldChange|> 1 and adjusted p-value (padj) <0.05. Differential splicing was detected with LeafCutter (version 0.2.9)[99] with the significance thresholds |deltapsi| >0.1 and adjusted p-value (p.adjust) <0.05. Differential transcript usage was computed with IsoformSwitchAnalyzeR (version 1.10.0) and the DEXSeq method[75,100–104]. Significance thresholds were |dIF| >0.1 and adjusted p-value (isoform_switch_q_value) <0.05.

PTC status of transcript isoforms with the annotated open reading frame was determined by IsoformSwitchAnalyzeR using the 50 nucleotides (nt) rule of NMD[75,105–107]. Isoforms with no annotated open reading frame in GENCODE were designated "NA" in the PTC analysis.

The control, SMG7, and SMG6/7 knockdown datasets (Gene Expression Omnibus, GEO: GSE86148)[14] were processed and analyzed with the same programs, program versions, and scripts as the SMG7 KO dataset, with minor changes due to the different sequencing method (paired-end vs. single-end). All scripts and parameters for the RNA-Seq analysis are available at GitHub [https://github.com/boehmv/SMG5-SMG7]. Overlaps of data sets were represented via nVenn[108] or the ComplexHeatmap package (version 2.6.2)[109]. Integrative Genomics Viewer (IGV) (version 2.8.12)[110] snapshots were generated from mapped reads (BAM files) converted to binary tiled data (tdf), using Alfred[111] with resolution set to 1 and IGVtools. Mean junction counts were obtained from sashimi plots generated using ggsashimi[112].

**Protein structure modeling and visualization**. The structure of human SMG7 (PDB: 1YA0) was superimposed onto the *C. elegans* SMG7 (PDB: 3ZHE) using the MatchMaker command in Chimera version 1.13[113], to generate a hsSMG7-ceSMG5 hybrid model. ChimeraX version 1.0[114] was used to visualize the modeled structure.

**Northern blotting**. The cells were harvested in peqGOLD TriFast reagent (VWR) and total RNA extraction was performed as described above. 2–4 µg of total RNA were resolved on a 1% agarose/0.4 M formaldehyde gel using the tricine/triethanolamine buffer system[115] followed by a transfer on a nylon membrane (Roth) in

10× SSC. The blots were incubated overnight at 65 °C in Church buffer containing α-32P-GTP [800 Ci/mmol, 10 mCi/ml] body-labeled RNA probes for detection of the reporter mRNA[80].

Endogenous 7SL RNA was detected by a 5′-32P-labeled oligonucleotide (5′-TG CTCCGTTTCCGACCTGGGCCGGTTCACCCCTCCTT-3′) for which γ-32P-ATP [800 Ci/mmol, 10 mCi/ml] was used for labeling. For NOP56 northern blots, the ex8b riboprobe sequence (5′-GAAACUUGGUCCCUUUGCUGGGCCCUGG GAAUCACUCAGACACCAGGACUGGCCAUCACCCCCAUAGCAGAGGCC UGUAUAGGUCAGGGAGCCCUGGUCAGCCAUCACCGUGAUCCCCAAC AAGCAGUGGGCACCAGAAGUGGCACCUGAUU-3′)[81] was cloned into the pSP73 vector, linearized and in vitro transcribed using α-32P-GTP [3000 Ci/mmol, 10 mCi/ml]. Ethidium bromide-stained 28S and 18S rRNA served as loading controls. RNA signal from at least three distinct samples was detected with the Typhoon FLA 7000 (GE Healthcare) and was quantified in a semi-automated manner using the ImageQuant TL 1D software (version 8.1) with a rolling-ball background correction. EtBr-stained rRNA bands were quantified with the Image Lab 6.0.1 software (Bio-Rad). Signal intensities were normalized to the internal control (7SL or rRNA) before the calculation of mean values. The control condition was set to unity (TPI WT for reporter assays), quantification results are shown as data points and mean.

**RNA immunoprecipitation (RIP).** The RIP protocol was adapted from Lee et al.[48]. Two days after seeding and reverse transfecting cells with siRNA, the cells were washed with 2 ml PBS, harvested in 1 ml PBS, collected for 10 min at $100 \times g$, and resuspended in 300 µl RIP lysis buffer (10 mM Tris pH 7.5, 150 mM NaCl, 2 mM EDTA, 0.1% Triton X-100) supplemented with 1 tablet of PhosSTOP (Roche), 100 µl EDTA-free HALT Protease and Phosphatase Inhibitor (ThermoFisher) and 20 µl RNasin (Promega) per 10 ml buffer. Protein concentration was measured, adjusted to 1 mg total protein in 450 µl RIP buffer, 20 µl of the sample was added to 500 µl peqGOLD TriFast and saved as input sample. The remaining sample was divided into 2× 200 µl and combined with 30 µl pre-washed Dynabeads Protein G beads (ThermoFisher), which were pre-incubated with 5 µg of either purified goat IgG (control; Bethyl, P50-100) or UPF1 antibody (Bethyl, A300-036A). The samples were incubated in an overhead rotator for 1 h at 4 °C, washed 5× with 1 ml RIP Wash Buffer (5 mM Tris pH 7.5, 150 mM NaCl, 0.1% Triton X-100), and co-immunoprecipitated RNA was recovered by incubating the beads with 500 µl RNA-solv. for 10 min at room temperature. Both input and IP samples were subjected to RNA extraction with the addition of 1 µl Precipitation Carrier. Ten microliters of resuspended RNA were used for reverse transcription and 2% of cDNA was used for quantitative PCR.

**Co-immunoprecipitation.** FLAG-tagged proteins were expressed in stable cell lines ($2.5-3.0 \times 10^6$ cells per 10 cm dish) induced for 48–72 h. The samples were lysed in 600 µl buffer E (20 mM HEPES-KOH (pH 7.9), 100 mM KCl, 10% glycerol, 1 mM DTT, Protease Inhibitor, 1 µg ml$^{-1}$ RNase A) and sonicated using the Bandelin Sonopuls mini20 with 15 pulses (2.5 mm tip, 1 s pulse, 50% amplitude). Concentration-adjusted lysates were subjected to immunoprecipitation for 2 h at 4 °C with overhead shaking using Anti-FLAG M2 Magnetic Beads (Sigma-Aldrich), the beads were washed three times for 5 min with buffer E, mild wash buffer (20 mM HEPES-KOH (pH 7.9), 137 mM NaCl, 2 mM MgCl₂, 0.2% Triton X-100, 0.1% NP-40) or medium wash buffer (20 mM HEPES-KOH (pH 7.9), 200 mM NaCl, 2 mM MgCl₂, 0.2% Triton X-100, 0.1% NP-40, 0.05% Na-deoxycholate). Co-immunoprecipitated proteins were eluted with SDS-sample buffer, separated by SDS-PAGE, and analyzed by immunoblotting.

**Label-free quantitative mass spectrometry.** FLAG-tagged SMG5 proteins or FLAG control were expressed in stable cell lines ($2.5-3.0 \times 10^6$ cells per 10 cm dish) induced with 1 µg ml$^{-1}$ doxycycline for 72 h. Samples were lysed and immuno-precipitated as described above, using mild wash buffer for washing steps and eluted in 42.5 µl of a 200 mg ml$^{-1}$ dilution of FLAG peptides (Sigma) in 1× TBS. 1 volume of 10% SDS was added and the samples were reduced with DTT and alkylated with CAA (final concentrations 5 mM and 55 mM, respectively). Tryptic protein digestion was performed using a modified version of the single pot solid phase-enhanced sample preparation (SP3)[116]. In brief, reduced and alkylated proteins were supplemented with paramagnetic Sera-Mag speed beads (Thermo Fisher Scientific) and mixed in a 1:1-ratio with 100% acetonitrile (ACN). After 8 min incubation protein-beads-complexes were captured using an in-house build magnetic rack and two times washed with 70% EtOH. Afterward, samples were washed once with 100% ACN, air-dried and reconstituted in 5 µl 50 mM Trie-thylammonium bicarbonate supplemented with 0.5 µg trypsin and 0.5 µg LysC and incubated overnight at 37 °C. On the next day, the beads were resuspended and mixed with 200 µl ACN, incubated for 8 min, and again placed on the magnetic rack. Tryptic peptides were washed once with 100% ACN, airdried, dissolved in 4% DMSO, and transferred into 96-well PCR tubes. After acidification with 1 µl of 10% formic acid, the samples were ready for LC-MS/MS analysis.

Proteomics analysis was performed by data-dependent acquisition using an Easy nLC1000 ultra-high-performance liquid chromatography (UHPLC) system coupled via nanoelectrospray ionization to a Q Exactive Plus instrument (all Thermo Scientific). Tryptic peptides were separated based on their hydrophobicity using a chromatographic gradient of 45 min (affinity enrichments samples) or 60 min (TurboID samples) with a binary system of buffer A (0.1% formic acid) and buffer B (80% ACN, 0.1% formic acid). In-house-made analytical columns (length: 50 cm, inner diameter: 75 µm) filled with 1.9 µm C18-AQ Reprosil Pur beads (Dr. Maisch) were used for separation. Using the 45 min chromatographic gradient, buffer B was linearly increased from 9 to 30% over 30 min followed by a steeper increase to 47% within 6 min. Finally, buffer B was increased to 95% within 4 min and stayed at 95% for 5 min to wash the analytical column. For the 60 min chromatographic gradient, buffer B was linearly increased from 3 to 30% over 40 min followed by an increase to 50% within 8 min. Finally, buffer B was increased to 95% within 1 min and the column washed for 10 at 95%. Full MS spectra (300–1750 $m/z$) were acquired with a resolution of 70,000, a maximum injection time of 20 ms, and an AGC target of 3e6. The top 10 most abundant peptide ions of each full MS spectrum were selected for HCD fragmentation (NCE: 26) with an isolation width of 1.8 $m/z$ and a dynamic exclusion of 20 s. MS/MS spectra were measured with a resolution of 17,000, a maximum injection time of 60 ms and an AGC target of 5e5 for the 45 min gradient, and a resolution of 35,000, a max. injection time of 110 ms and an AGC target of 5e5 using the 60 min gradient.

MS RAW files were analyzed using the standard settings of the MaxQuant suite (version 1.6.17.0)[117]. Peptides were identified by matching against the human UniProt database using the Andromeda scoring algorithm[118]. Carbamidomethylation of cysteine was set as a fixed modification, methionine oxidation, and N-terminal acetylation as variable modification. Trypsin/P was selected as the digestion protein. A false discovery rate (FDR) <0.01 was used for the identification of peptide-spectrum matches and protein quantification. Intensities were calculated using the LFQ algorithm implemented in MaxQuant with the standard parameters. Data processing and statistical analysis were done in the Perseus software (version 1.6.5.0)[119]. LFQ-normalized protein intensities were Log2-transformed for normal distribution and filtered for at least three valid values in at least one sample group. Missing values were imputed by drawing random values from a 1.8 standard deviations downshifted, 0.3 standard deviations broad normal distribution. Statistical testing was performed using a two-sided Welch's $t$-test with permutation-based FDR correction (FDR = 0.05, S0 = 0.1). Significantly different proteins were identified using the following cut-off: $q$-value <0.05, absolute log2 fold change >1. Visualization was performed with InstantClue[120], the R (version 4.0.4) package ggplot2 (version 3.3.3) or ComplexHeatmap (version 2.6.2)[109].

**TurboID proximity labeling.** Stable WT or SMG7 KO cell lines were seeded ($5 \times 10^6$ cells per 10 cm dish) and reverse transfected using 6.25 µl Lipofectamine RNAiMAX and 200 pmol siRNA (control or SMG5). The expression of FLAG-TurboID-tagged UPF1 or control proteins was induced on the next day with 30 µg ml$^{-1}$ cumate. In this step the medium was also changed to high-glucose, GlutaMAX DMEM (Gibco) supplemented with 9% dialyzed fetal bovine serum (Gibco; A3382001; to suppress background biotinylation)[121] and 1× Penicillin Streptomycin (Gibco). Proximity labeling by biotinylation was performed on the next day by the addition of 50 µM biotin for 15 min. Afterward, the cells were washed twice with PBS on ice, scraped in 1 ml ice-cold PBS, collected for 5 min at $100 \times g$ and 4 °C, and finally resuspended in 200 µl phospho-RIPA buffer (50 mM Tris pH 8.0, 150 mM NaCl, 1% IGEPAL CA-630, 0.5% deoxycholate, 0.1% SDS, 1 µg ml$^{-1}$ RNase A) supplemented with 1 tablet of PhosSTOP (Roche), 100 µl EDTA-free HALT Protease and Phosphatase Inhibitor (ThermoFisher) and per 10 ml buffer. Samples were sonicated using the Bandelin Sonopuls mini20 with 10 pulses (2.5 mm tip, 1 s pulse, 50% amplitude). Fifty microliter input aliquots containing 100 µg of total protein were prepared and mixed with SDS-sample buffer. Concentration-adjusted lysates containing 1 mg total protein in 500 µl buffer were concentrated to approximately 100 µl in 0.5 ml Amicon Ultra centrifugal filter devices (3K cutoff) for 45 min at 4 °C and 14.000 $\times g$, to minimize excess biotin in the sample. The concentrated sample was combined with 200 µl RIPA buffer (wash of centrifugal filter), mixed with 25 µl pre-washed Pierce Streptavidin Magnetic Beads (ThermoFisher), and incubated for 2 h at 4 °C with overhead shaking. The beads were washed four times for 5 min with 800 µl RIPA buffer, followed by one wash with 800 µl mild wash buffer (20 mM HEPES-KOH (pH 7.9), 137 mM NaCl, 2 mM MgCl₂, 0.2% Triton X-100, 0.1% NP-40). Biotinylated proteins were eluted with 50 µl 1× SDS-sample buffer, supplemented with 20 mM DTT and 3 mM biotin, for 15 min at 96 °C, followed by another elution with 25 µl and both eluates were combined. Aliquots (10 µl input, 12.5 µl eluate) of the samples were resolved on 10% polyacrylamide gels containing 25 µl TCE (2,2,2-trichloroethanol) to allow fluorescent visible detection of proteins[122] and subsequently used for western blotting. Tryptic protein digestion and proteomics analysis were performed as described above. Of note, no SMG5 Label-Free Quantification (LFQ) intensities were obtained, although SMG5 peptides were only detected in the WT samples and not in the SMG7 KO.

**Data presentation.** Quantifications and calculations for other experiments were performed—if not indicated otherwise—with Microsoft Excel (version 1808) or R (version 4.0.4) and all plots were generated using IGV (version 2.8.12), GraphPad Prism 5, ggplot2 (version 3.3.3), or ComplexHeatmap (version 2.6.2). Boxplots were generated using the geom_boxplot() function of ggplot2 with the centerline representing the 50th percentile (median), whereas the lower and upper box limits correspond to the 25th and 75th percentiles. The whiskers extend from the box

limits to the smallest or largest value no further than 1.5 * inter-quartile range. Data beyond the end of the whiskers are plotted individually.

**Reporting summary**. Further information on research design is available in the Nature Research Reporting Summary linked to this article.

## Data availability

RNA-sequencing data generated for this manuscript have been deposited in the ArrayExpress database at EMBL-EBI (www.ebi.ac.uk/arrayexpress)[123] under accession number E-MTAB-9330. Published datasets analyzed for this paper include Gene Expression Omnibus (GEO) accession number GSE86148. Data of human proteome abundances were retrieved from https://proteomesoflife.org/ with the human taxonomy identifier (9606) on 2021-03-03 (03. March 2021)[89]. The mass spectrometry proteomics data have been deposited to the ProteomeXchange Consortium via the PRIDE[124] partner repository with the dataset identifier PXD024747. Published protein structures were human SMG7 (PDB: 1YA0 [https://doi.org/10.2210/pdb1YA0/pdb]) and *C. elegans* SMG5-SMG7 (PDB: 3ZHE [https://doi.org/10.2210/pdb3ZHE/pdb]). All relevant data supporting the key findings of this study are available within the article and its Supplementary Information files or from the corresponding author upon reasonable request. Source data—where applicable—are provided for all figures, including raw images of EtBr-stained agarose gels, western blots, and northern blots, as well as qPCR raw values, quantification, and an overview file stating all further necessary information (e.g., which antibody was used). All raw source data can also be accessed at Zenodo [https://doi.org/10.5281/zenodo.4603278]. Source data are provided with this paper.

## Code availability

The codes used in this study are available at GitHub (https://github.com/boehmv/SMG5-SMG7) and Zenodo (https://doi.org/10.5281/zenodo.4603388).

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

## Acknowledgements
We thank members of the Gehring lab for discussions and reading of the manuscript. The authors thank Sarah Gerlich and Jan Riemer (Institute of Biochemistry, University of Cologne) for their help with the TurboID experiments. We also thank Marek Franitza and Christian Becker (Cologne Center for Genomics, CCG) for preparing the sequencing libraries and operating the sequencer. We acknowledge Tobias Jakobi for helping with infrastructure support. This work was supported by grants from the Deutsche For-schungsgemeinschaft to C.D. (DI 1501/8-1, DI1501/8-2) and N.H.G (GE 2014/6-2 and GE 2014/10-1). V.B. was funded under the Institutional Strategy of the University of Cologne within the German Excellence Initiative. N.H.G. acknowledges support by a Heisenberg professorship (GE 2014/7-1 and GE 2014/13-1) from the Deutsche For-schungsgemeinschaft. C.D. and T.B.B. were kindly supported by the Klaus Tschira Stiftung gGmbH (00.219.2013). This work was supported by the DFG Research Infra-structure as part of the Next Generation Sequencing Competence Network (project 423957469). NGS analyses were carried out at the production site WGGC Cologne.

## Author contributions
Conceptualization: NH.G., S.K., and V.B. Methodology: V.B., S.K., N.H.G. Software: V.B., T.B.-Bo., J.V.G., S.K., and C.D. Investigation: S.K., V.B., J.V.G., and S.K. Resources and data curation: V.B., T.B-B., J.A., S.K., M.K., and C.D. Writing—original draft, review, and editing: V.B., J.V.G., N.H.G., and S.K. Visualization: V.B., S.K., and J.V.G. Supervision: N. H.G. and V.B. Funding acquisition: N.H.G. and C.D.

## Funding

## Competing interests
The authors declare no competing interests.
