## [Peer Review File · Nature Communications]

REVIEWER COMMENTS

Reviewer #1 (Remarks to the Author):

The manuscript entitled "Nonsense-mediated mRNA decay relies on "two-factor authentication by SMG5-SMG7" by Boehm, Kueckelmann et al. addresses the mechanism of target recognition in the NMD pathway by the SMG5, SMG6 and SMG7 proteins, together with phosphorylated UPF1. The prevailing model of NMD in metazoans suggests that the resident time of UPF1 on an mRNA transcript is an indicator of NMD and that hyperphosphorylation of UPF1 and its subsequent recognition by SMG5, SMG7 and SMG6 commits a UPF1-bound mRNA to decay through NMD. The proteins SMG5 and SMG7 form a heterodimer, which was earlier shown to bridge the NMD machinery to the CCR4-NOT deadenylation complex. SMG6, on the other hand, possesses endonucleolytic activity and plays an important role in NMD. These proteins were shown to interact with phosphorylated UPF1 through their TPR domains, although SMG6 was also shown to interact with UPF1 in a phosphorylation-independent manner. Based on knockdown and rescue studies, the roles of SMG5-7 and SMG6 were thought to be interchangeable, owing to a functional redundancy among the proteins.

In this study, the authors generated a SMG7 KO cell-line and used this tool to investigate the mechanism of action of SMG7, SMG5 and SMG6 in further detail. They found, using RT-PCR assays on specific targets as well as RNA-Seq, that deletion of SMG7 significantly abolished NMD and leads to upregulation of several transcripts. This is in contrast to previous studies with siRNA-mediated SMG7 knockdown where the effects were much milder. At that time, this was presumed to be due to compensatory effect of SMG6. Interestingly siRNA KD of SMG5 in the background of SMG7 KO further heightened the effect on NMD. The effect on NMD could be rescued by addition of wildtype SMG7 and a SMG7 mutant that is incapable of binding phospho-UPF1, but not one that is incapable of binding SMG5. Furthermore, a SMG7 construct lacking the C-terminus (which was proposed to recruit the deadenylation machinery) is capable of rescuing NMD. These new data cast a doubt on the hitherto proposed function of SMG7. SMG5 was largely capable of rescuing the effect of SMG7 KO, though surprisingly the catalytically inactive PIN domain was found to be essential to recapitulate this effect. Depletion of SMG5 together with a complete KO of SMG7 also affected the endonucleolytic activity of SMG6.

The data presented in this paper are convincing and presented very clearly. The Methods section is sufficiently detailed. The extended data included also support the figures in the main text.

Major Points:

- It would be very helpful if the authors could indicate the extent of knockdown in each of the siRNA treated samples (in terms of residual protein or RNA) as it gives the reader a feeling for how strong the effects of knockdown of SMG5 and SMG6 are.
- From the data presented here, it is clear that there is a functional dependency of SMG6 on SMG5 and SMG7. However, the hierarchy in terms of action of the proteins is not convincing. It appears that at any given point of time in the cell, a combination of two of the three proteins (SMG5/6/7) is essential to mediate NMD. It is possible that the main scaffolding protein is SMG7, which is why knockout of SMG7 has a strong effect. It must be noted (as shown in Figure 2D) that KD of SMG6 in SMG7-KO cells also has a fairly strong effect, similar to that of SMG5. Although the TPR domain of SMG6 does not interact with SMG5 or SMG7, it would be worthwhile to test if full-length SMG6 interacts with either SMG7 or SMG5, and if this interaction is necessary for stability or function of SMG6. As such, from this manuscript it is not clear why SMG5-7 should be necessary for SMG6 activity.
- The authors speculate that binding of SMG5-7 to phosphorylated UPF1 induces a conformational change within UPF1, which might facilitate binding of SMG6. However, there is no evidence to support this statement, which is an important step in the "two-step authentication model". This would require careful analysis of binding of SMG6 to UPF1 in the presence and absence of SMG5-7 and upon different phosphorylation of UPF1 to different extents, which might be out of the scope of this

manuscript.

- Furthermore, establishing a hierarchy in the NMD pathway calls for demonstrating that depletion of SMG6 does not impact the activity of SMG5-7 at all. KD of SMG6 does in fact show a slight increase of the target transcripts, as shown in Figure 2D.

A minor point in addition: the last sentence in the figure legend of 1g-i is repeated (Density plots show the distribution...)

In summary, I suggest revising the discussion of the manuscript to focus only on the functional redundancy of SMG5, SMG7 and SMG6, which on its own is fairly interesting (albeit to NMD aficionados).

Reviewer #2 (Remarks to the Author):

This study from the Gehring lab provides convincing evidence against the currently popular idea in the NMD field positing that in mammalian cells, degradation of NMD-targeted mRNAs can be initiated independently by either of two pathways, namely by SMG6-mediated endonucleolytic cleavage near the stop codon or by SMG5-SMG7-mediated recruitment of the CCR4-NOT deadenylase complex. Boehm and colleagues now show that the combined loss of SMG5 and SMG7 completely inactivates NMD. Their findings are consistent with another recently published preprint by the Leeb lab (Galimberti et al., bioRxiv 10.1101/2020.07.07.180133v1), so there is no doubt anymore that SMG6 activity requires the presence of at least one of the two heterodimerizing factors, SMG5 or SMG7. Based on this data, the authors propose a “two-factor authentication” model to activate SMG6-mediated mRNA cleavage, in which hyperphosphorylation represents the first factor and SMG5-SMG7 recruitment the second factor. Moreover, the authors report the surprising finding that SMG5 can functionally substitute for SMG7 and vice versa, but it remains unclear what exact function in the NMD pathway both of these proteins are able to execute in the absence of the other. It would be highly insightful to check if SMG5 and SMG7 form homodimers if the normal binding partner is absent.

HEK cells with a SMG7 knockout were generated, which proliferated a bit slower than the parental cells but showed no decrease in viability (Fig. 1). While WT SMG7 and a 14-3-3 mutant could rescue the NMD deficiency, the G100E mutant could not (Fig. 2), suggesting that SMG7 needs to interact with SMG5 for NMD activity. Interestingly, siRNA-mediated knockdowns of SMG5, SMG6 or both together in the SMG7 KO cells led to a much stronger, apparently complete NMD inhibition, indicating a synergistic function between SMG7 and SMG5 and between SMG7 and SMG6. Interestingly, RNA-seq of these conditions with complete NMD inactivation revealed that approx. 40% of the expressed genome of the HEK cells is under direct or indirect control of NMD (Fig. 3), which is much more than previously estimated based on incomplete NMD inhibition. Following up on the indication that SMG6-mediated endonucleolytic cleavage might require SMG5 and SMG7, TPI reporters with an XRN1-resistant pseudoknot structure were used and indeed, when the SMG7 KO was combined with SMG5 KD, no endonucleolytic cleavage was observed anymore. Interestingly, SMG7 KO alone or SMG5 KD alone did not significantly inhibit the endonucleolytic cleavage.

The authors then went on to test additional SMG7 mutants for their ability to rescue NMD in the SMG7 KO cells, combined or not with SMG5 or SMG6 KDs. Here is where the manuscript loses focus. This reviewer finds that the reported data raises more questions than it answers and hence suggests to removing that data and instead use it as the starting point for a follow-up manuscript that focuses on the different mechanistic possibilities to decay NMD-sensitive mRNAs. The rescue experiments need to be complemented with other approaches, including IPs in order to become fully informative. Finally, and intriguingly, the authors also found that overexpression of SMG5 in the SMG7 KO cells can rescue the diminished NMD activity, indicating once more that the previously reported SMG5-SMG7 heterodimer formation is not required for NMD. As for Fig. 5, I do not find the rescue experiments with various SMG5 mutants very informative in the absence of complementing approaches and more work

would be needed to generate compelling new insights. Among several open questions, it should for example be tested whether the capability of SMG5 to sustain NMD in the absence of SMG7 might require PNRC2. Again, I suggest to spare such additional investigations for a follow-up manuscript instead of showing it here, as the data in its current form rather distracts and confuses the reader. The Discussion is rather long and attempts to unite the existing NMD literature and their data presented herein into a coherent NMD working model that they call the “two-factor authentication” model. I think this exercise is well done and resulted in an admittedly quite speculative but nevertheless attractive Fig. 7 illustrating that model. It is eventually an editorial decision how much speculative assumptions should be tolerated in working model figures, but this reviewer likes the model and sees its value in that it provides many ideas for follow-up experiments to test specific assumptions of the model. In that sense, the model serves its purpose.

Main points to address:

- For Fig. 2, immunoprecipitations should be performed to confirm that 14-3-3mut indeed fails to interact with p-UPF1 and that G100E fails to interact with SMG5.
- The experiment shown in Fig. 4 lacks the SMG6 KD as a control to show that generation of the observed xrFRAG and 3' fragment depends on SMG6 activity. It is assumed, probably based on experiments of that type in a previous publication, but it should be added here again for completeness. Otherwise the conclusion that the SMG5-SMG7 heterodimer is required for SMG6 activity is not warranted.
- Since The authors report evidence that SMG5 and SMG7 can functionally substitute for each other, it should be tested in Fig. 4e-j if SMG5 KD in WT HEK cells also leads to increased UPF1 phosphorylation and accordingly increased IP of the known NMD-targeted transcripts with hyperphosphorylated UPF1.
- Fig. 5 provides a lot of data, but conclusions are difficult because of lacking controls or complementary experiments. To solidify the conclusions drawn from the NMD rescue capacity of the tested SMG7 mutants, IPs should be done to validate that e.g. the 1-633 mutant indeed fails to interact with POP2. Additionally, testing these IPs for co-precipitation of PP2A might be informative. Moreover, the finding that in the SMG7 KO / SMG6 KD condition, expression of the truncated SMG7 1-633 construct can rescue NMD begs the question how RNA degradation is triggered in this case. Is the residual SMG6 still sufficient to trigger endonucleolysis or is another pathway taking over, e.g. deadenylation-independent decapping? Collectively, I think that these experiments raise more questions than they answer and would need additional investigation. Rather than including them in this anyway already data-heavy manuscript, they might serve instead as the starting point for another manuscript focusing on the different mechanisms to degrade NMD-sensitive mRNAs.
- The comment made for Fig. 5 is also valid for Fig. 6: showing that SMG5 can substitute for SMG7 in NMD is compelling, but the rescue experiments need to be complemented with additional experiments in order to be conclusive and hence they should be spared for a follow-up manuscript. Panel 6c (northern blot with Nop65) could be moved to Fig. 4 and replace the current panel 4c there.

Minor points:

Line 69: The first NMD factors were discovered about 30 years ago, thus “several decades” might be more appropriate than “many decades”.

Fig. 1b and Ext Fig. 1: please indicated what the bands are that are depicted by a *

(This report is from Oliver Mühlemann)

Reviewer #3 (Remarks to the Author):

Nonsense-mediated mRNA decay relies on “two-factor authentication” by SMG5-SMG7

Volker Boehm et al.

In this recent work, Gehring and colleagues investigate the role of SMG7 on targeting mRNA to nonsense-mediated RNA decay. Cumulative data from a number of labs have implicated the SMG5-SMG7 heterodimer in binding the core NMD factor, UPF1, and recruiting the CCR4/NOT deadenylation complex to NMD substrates to promote deadenylation and exonucleolytic decay of mRNA targets.

CRISPR knock-out 293T clones lacking detectable SMG7 were generated; these cells demonstrated slower proliferation and increased levels of two characterized NMD transcripts (as monitored by end-point RT-PCR). RNA-Seq analysis of these clones identified global gene expression changes with NMD-sensitive mRNA isoforms constituting a significant fraction of up-regulated transcripts, as expected. Complementation assays were used to demonstrate that both wild-type and a SMG7 mutant lacking the ability to interact with UPF1 rescue the NMD defect in SMG7 KO cells, whereas a SMG7 mutant unable to interact with SMG5 did not. In experiments in which additional NMD components were knocked down in SMG7 KO cells, depletion of the SMG6 endonuclease was shown to completely abolish NMD, as anticipated when both exonucleolytic and endonucleolytic pathways for targeting NMD substrates to decay are eliminated. Surprisingly, KD of SMG5 in SMG7 KO cells also completely abolished NMD, suggesting that the two decay pathways are not independent as previously assumed. Consistent with this observation, SMG6-mediated endonucleolytic cleavage of both an NMD reporter and the endogenous NMD target, NOP54, were disrupted in cells lacking SMG5-SMG7. Interestingly, while the level of phospho-UPF1 increases in SMG7 KO cells and UPF1 is found to show increased interaction with non-target mRNAs, these increases were not further increased with SMG5 KD (where the largest impact on NMD is observed) and thus are unlikely to underlie NMD inhibition in SMG7/SMG5 KO/KD cells. Moreover, SMG7 deletion mutants unable to interact with the CCR4/NOT deadenylase were able to restore NMD in SMG7 KO, SMG7/SMG5 and SMG7/SMG6 KO/KD cells, indicating a novel role for SMG7 in NMD. Finally, using SMG7 KO/SMG5 KD cells, the authors show that SMG5 over-expression can compensate for loss of SMG7 and that both its 14-3-3 domain (involved in binding UPF1) and the C-terminus (thought to promote dephosphorylation of UPF1) are required for SMG5 activity in NMD (including SMG6-mediated substrate cleavage).

The data provide novel and convincing evidence for the dependence of SMG5/7 in SMG6-mediated endonucleolytic decay of NMD substrates and questions the role for recruitment of the CCR4/NOT deadenylase in degrading targets. Moreover, the observation that SMG7 deficient in CCR4/NOT binding can restore NMD in cells lacking SMG6 suggests a novel, yet undetermined role for SMG5/7 in NMD. The work falls significantly short in an effort to understand how SMG5/7 are required for SMG6 function (likely through mediating the interaction between UPF1 and SMG6, but untested) or the curious new role for the SMG5/7 proteins in NMD, and instead propose an over-arching model for how these three factors function in NMD based primarily on past literature.

1. The current study provides a multitude of novel and interesting findings from transcriptome analyses, complementation tests and reporter assays; however, the data is generally skimmed over by the authors and it is left to the reader to scrutinize the figures to find the nuggets of supporting data. While this is difficult in itself due to the presentation of the large datasets (and a vague description on how to interpret; i.e. Fig 1f and 3h), some of the most intriguing findings seem not to be fully highlighted - for example, in Fig 5b, the ability of a SMG7 mutant deficient in recruiting the CCR4/NOT deadenylase to complement the SMG7/SMG6 KO/KD cells provides clear evidence for a novel role of SMG7 in NMD outside of deadenylation or an ability to recruit/activate SMG6. Additionally, although data figures show replicates, averages and standard deviations, the text describing the data is highly qualitative.

2. Strong evidence is presented showing a requirement for SMG5/7 in SMG6-mediated NMD substrate cleavage. One obvious mechanism to explain this observation is through recruitment of SMG6 to UPF1 via SMG5/7, and this should be tested by monitoring SMG6-UPF1 interaction in the various cell lines.

3. The authors observe "a complete inhibition of NMD" upon SMG5 or SMG6 KD in SMG7 KO cells. While inhibition of NMD is clearly more pronounced in these cells than in SMG7 knockout alone (Fig

2d), a UPF1 KD control (the 'gold standard' in NMD) is needed to help gauge the extent of NMD inhibition in cells depleted of SMG7, SMG5/7 and SMG6/7.

4. Mutant alleles of SMG7 are expressed at different levels, and so a quantification of IP versus input should be provided for the coIP experiment (Fig 5c).

4. Ectopic expression of wild-type SMG5 or the G120E mutant (unable to bind SMG7) is shown to complement SMG7 KO (Fig 6b). It needs to be clarified that SMG5 is over-expressed in these cells and some indication of its level over endogenous SMG5 (and compare to endogenous SMG7) provided. It is reasonable to assume that SMG5 can form homodimers in this context that function in NMD similarly to SMG5/7 heterodimers, however, this is not tested or discussed.

6. The authors convincingly show in Fig 1 that their SMG7 KO clones have stronger NMD defects than a SMG7 KO (either in their hands or based on previous data by Colombo et al using HeLa cells). This is not unexpected. Moreover, the detailed comparison of transcriptome changes between the SMG7 KO in 293T cells and SMG7 KD in HeLa is comparing apples-to-oranges and distracts from other key findings in the paper.

7. The discussion section of the manuscript and model figure presented in Fig 7 provides a comprehensive overview of NMD events based primarily on published literature and is better suited for a review article. Critically, the authors fail to hone-in on the novel findings gained from their study and how these observations should change how we think about events occurring between (presumably) the NMD machinery and terminating ribosome. There are many provocative findings from this study and the impact of their data and novel roles for SMG5 and SMG7 in eliciting NMD should be the focus of the discussion and better reflected in their model figure(s) and in the manuscript title.

General response to the reviewers

We very much appreciate the interest of all three reviewers in our work and are grateful for their thorough evaluation of the manuscript and for their constructive criticisms. In the revised version we have addressed most of their comments by modifying the text and by including new experimental data as described in this detailed point-by-point response. Text changes made to the manuscript during the revision are tracked with Word's 'Track Changes'.

The highlights of the revised manuscript include:

- Restructured results part with condensed functional characterization of SMG5 and SMG7 to improve readability.
- Various additional aspects addressed, e.g. the role of PNR2 for SMG5 activity.
- New interaction data using SMG5-SMG7 co-IPs, UPF1 Turbo-ID proximity labelling and mass spectrometry supporting the "two-factor authentication" model.
- Substantially revised discussion and model.

The new mass spectrometry proteomics data are available via ProteomeXchange with identifier PXD024747 and the following reviewer account details:

- Username: reviewer_pxd024747@ebi.ac.uk
- Password: nnmcp8m8

REVIEWER COMMENTS

Reviewer #1 (Remarks to the Author):

The manuscript entitled "Nonsense-mediated mRNA decay relies on "two-factor authentication by SMG5-SMG7" by Boehm, Kueckelmann et al. addresses the mechanism of target recognition in the NMD pathway by the SMG5, SMG6 and SMG7 proteins, together with phosphorylated UPF1. The prevailing model of NMD in metazoans suggests that the resident time of UPF1 on an mRNA transcript is an indicator of NMD and that hyperphosphorylation of UPF1 and its subsequent recognition by SMG5, SMG7 and SMG6 commits a UPF1-bound mRNA to decay through NMD. The proteins SMG5 and SMG7 form a heterodimer, which was earlier shown to bridge the NMD machinery to the CCR4-NOT deadenylation complex. SMG6, on the other hand, possesses endonucleolytic activity and plays an important role in NMD. These proteins were shown to interact with phosphorylated UPF1 through their TPR domains, although SMG6 was also shown to interact with UPF1 in a phosphorylation-independent manner. Based on knockdown and rescue studies, the roles of SMG5-7 and SMG6 were thought to be interchangeable, owing to a functional redundancy among the proteins.

In this study, the authors generated a SMG7 KO cell-line and used this tool to investigate the mechanism of action of SMG7, SMG5 and SMG6 in further detail. They found, using RT-PCR assays on specific targets as well as RNA-Seq, that deletion of SMG7 significantly abolished NMD and leads to upregulation of several transcripts. This is in contrast to previous studies with siRNA-mediated SMG7 knockdown where the effects were much milder. At that time, this was presumed to be due to compensatory effect of SMG6. Interestingly siRNA KD of SMG5 in the background of SMG7 KO further heightened the effect on NMD. The effect on NMD could be rescued by addition of wildtype SMG7 and a SMG7 mutant that is incapable of binding phospho-UPF1, but not one that is incapable

of binding SMG5. Furthermore, a SMG7 construct lacking the C-terminus (which was proposed to recruit the deadenylation machinery) is capable of rescuing NMD. These new data cast a doubt on the hitherto proposed function of SMG7.

SMG5 was largely capable of rescuing the effect of SMG7 KO, though surprisingly the catalytically inactive PIN domain was found to be essential to recapitulate this effect. Depletion of SMG5 together with a complete KO of SMG7 also affected the endonucleolytic activity of SMG6.

The data presented in this paper are convincing and presented very clearly. The Methods section is sufficiently detailed. The extended data included also support the figures in the main text.

- We thank the reviewer for the overall positive feedback on our manuscript.

Major Points:

- It would be very helpful if the authors could indicate the extent of knockdown in each of the siRNA treated samples (in terms of residual protein or RNA) as it gives the reader a feeling for how strong the effects of knockdown of SMG5 and SMG6 are.
 - This is a good suggestion because the strength of the knockdown is likely to be critical for the strength of the expected effects. We have included exemplary westerns blots as Extended Data Fig. 3a, showing the residual amount of SMG5, SMG6 and SMG7 proteins after the respective knockdowns. A dilution series of the control sample was included to allow the reader to estimate the KD efficiencies. Furthermore, Supplementary Table 1 provides information on the RNA level about the knockdown of SMG5 and SMG6 in the RNA-seq data. A differential gene expression analysis of all NMD factors in the RNA-seq data is now also displayed in a comprehensive heatmap in Extended Data Fig. 5a.
- From the data presented here, it is clear that there is a functional dependency of SMG6 on SMG5 and SMG7. However, the hierarchy in terms of action of the proteins is not convincing. It appears that at any given point of time in the cell, a combination of two of the three proteins (SMG5/6/7) is essential to mediate NMD. It is possible that the main scaffolding protein is SMG7, which is why knockout of SMG7 has a strong effect. It must be noted (as shown in Figure 2D) that KD of SMG6 in SMG7-KO cells also has a fairly strong effect, similar to that of SMG5. Although the TPR domain of SMG6 does not interact with SMG5 or SMG7, it would be worthwhile to test if full-length SMG6 interacts with either SMG7 or SMG5, and if this interaction is necessary for stability or function of SMG6. As such, from this manuscript it is not clear why SMG5-7 should be necessary for SMG6 activity.
 - We agree that the reviewer's hypothesis that "two out of the three SMG5/6/7 proteins might be enough to mediate NMD" is very suggestive. However, we did not see any additive effects of the combined SMG5 + SMG6 KD over isolated SMG6 depletion (Fig. 2e and Extended Data Fig. 3b), which one would expect if the hypothesis was true.
 - To test if indeed SMG5 or SMG7 could interact with full-length SMG6, we performed several experiments. Standard FLAG-IPs with FLAG-tagged SMG7 or SMG5 did not show any detectable interaction with endogenous SMG6. Furthermore, we have performed label-free mass spectrometry analyses of the SMG5 interactome and could not detect any co-immunoprecipitated SMG6 (Supplementary Table 4). Furthermore, our new UPF1 TurboID proximity labelling data now shed light on the function of SMG5-SMG7 for NMD in general and the activity of SMG6 in particular (Fig. 6, Extended Data Fig. 7 and Supplementary Table 5). In the absence of SMG5-SMG7, we find that many NMD factors (including SMG6) are stronger biotinylated by TurboID-UPF1, which - in

combination with the functional data - corresponds in our view to the formation of inactive NMD complexes. Using these data, we developed several models that put our new findings in the context of NMD (see discussion and Fig. 7).

- The authors speculate that binding of SMG5-7 to phosphorylated UPF1 induces a conformational change within UPF1, which might facilitate binding of SMG6. However, there is no evidence to support this statement, which is an important step in the “two-step authentication model”. This would require careful analysis of binding of SMG6 to UPF1 in the presence and absence of SMG5-7 and upon different phosphorylation of UPF1 to different extents, which might be out of the scope of this manuscript.
 - We agree that crucial evidence supporting our speculation about SMG5-SMG7 potentially inducing a conformational change within UPF1 was missing from the previous version of the manuscript. Therefore, we have now removed most of this speculative aspect from the discussion as part of the revision.
 - However, the reviewers' questions about SMG6-UPF1 interaction prompted us to conduct further experiments. We have intensively tried to obtain reliable data showing whether any changes in the interaction between UPF1 and SMG6 occur in response to the presence or absence of SMG5 and/or SMG7. Standard IPs with either FLAG-tagged SMG6 or UPF1 were unsuccessful and did not consistently showed convincing binding to the other potential interaction partner over background controls. We have therefore switched to using the TurboID-mediated proximity labeling with the aim to capture transient interactions. To this end, we tried to express TurboID-tagged SMG6 and to check for transient interactions by proximity labeling, which failed due to the poor expression and therefore low labeling efficiency of this SMG6 construct. Conversely, TurboID-tagged UPF1 performed well, displayed convincing biotinylation of known interaction partners (e.g. UPF3B or STAU2). As outlined above, using TurboID-UPF1 we find that many NMD factors, including those responsible for the first authentication step, are stronger “proximity-labelled” in the absence of SMG5-SMG7. We suggest that this corresponds to the formation of SMG6-containing, inactive NMD complexes, which are at least temporarily arrested on NMD substrates. We feel that with this experiment and the resulting changes in the results section and discussion, we have now answered the reviewer's - admittedly legitimate - question.
- Furthermore, establishing a hierarchy in the NMD pathway calls for demonstrating that depletion of SMG6 does not impact the activity of SMG5-7 at all. KD of SMG6 does in fact show a slight increase of the target transcripts, as shown in Figure 2D.
 - We agree with the reviewer that the KD of SMG6 stabilizes many NMD targets, which is consistent with the hypothesis that SMG6-mediated endonucleolytic cleavage is the major degradation pathway of NMD. Moreover, SMG6 activity can be monitored by visualizing the degradation intermediates of the endonucleolytic cleavage (3' fragments). However, the degradative activity of SMG5-SMG7 is more difficult to study. While SMG7 has been reported to lead to accelerated deadenylation via the recruitment of the CCR4-NOT complex, we found that the C-terminal deletion mutant (which supposedly fails to interact with CNOT8/POP2, a component of the CCR4-NOT complex) fully rescued SMG7 KO. It is also highly peculiar that SMG5 - with no reported degradative ability - was able to rescue the SMG7 KO. As motivated by the comment of Oliver Mühlemann (see below), we also established that SMG5 does not rely on PNRC2 for NMD activity (Fig. 3c). Therefore, it is

currently unclear how to properly measure the activity of SMG5 or SMG7 and further research is needed to resolve this issue.

- Nevertheless, we understand the reviewer's objections to defining a strict hierarchy that would preclude the possibility that SMG6 also influences SMG5-7 activities. This aspect was not clearly discussed previously in the manuscript and we have now addressed this issue in the revised discussion. In summary, our data do not allow us to rule out the possibility that SMG6 has an effect on the activity of SMG5 or SMG7.

A minor point in addition: the last sentence in the figure legend of 1g-i is repeated (Density plots show the distribution...)

- This error was corrected.

In summary, I suggest revising the discussion of the manuscript to focus only on the functional redundancy of SMG5, SMG7 and SMG6, which on its own is fairly interesting (albeit to NMD aficionados).

- We took this concluding remark very seriously and restructured major parts of the manuscript and, in particular, rewrote the discussion. With these major revisions we wanted to increase the comprehensibility of the manuscript, to establish a more logical order of the experiments, and to focus the discussion on the most important findings of the work.

Reviewer #2 (Remarks to the Author):

This study from the Gehring lab provides convincing evidence against the currently popular idea in the NMD field positing that in mammalian cells, degradation of NMD-targeted mRNAs can be initiated independently by either of two pathways, namely by SMG6-mediated endonucleolytic cleavage near the stop codon or by SMG5-SMG7-mediated recruitment of the CCR4-NOT deadenylase complex. Boehm and colleagues now show that the combined loss of SMG5 and SMG7 completely inactivates NMD. Their findings are consistent with another recently published preprint by the Leeb lab (Galimberti et al., bioRxiv 10.1101/2020.07.07.180133v1), so there is no doubt anymore that SMG6 activity requires the presence of at least one of the two heterodimerizing factors, SMG5 or SMG7.

Based on this data, the authors propose a “two-factor authentication” model to activate SMG6-mediated mRNA cleavage, in which hyperphosphorylation represents the first factor and SMG5-SMG7 recruitment the second factor. Moreover, the authors report the surprising finding that SMG5 can functionally substitute for SMG7 and vice versa, but it remains unclear what exact function in the NMD pathway both of these proteins are able to execute in the absence of the other. It would be highly insightful to check if SMG5 and SMG7 form homodimers if the normal binding partner is absent.

- We thank Oliver Mühlemann for this suggestion. For us, the formation of SMG5-SMG5 or SMG7-SMG7 homodimers also seemed a logical explanation for the observed rescue effects in SMG5-SMG7. Accordingly, we investigated homodimerization of both SMG5 and SMG7 by immunoprecipitation from cell lysates, expressing simultaneously both FLAG- and clover-tagged versions of either SMG5 or SMG7. We used comparable conditions to those that were used for the rescue assays, because we wanted to avoid studying homodimerization in vitro, which may not occur in living cells. We used GST as a background control and the interaction between the “true” partner SMG5 or SMG7 as a positive control. In addition, we also used the interaction-defective mutants of SMG5 and SMG7, respectively, to learn more about the molecular details of the potential homodimers. However, neither SMG5 nor SMG7 appeared to form homodimers that could be detected with our experimental approach (see Figure below). We cannot rule out the possibility that homodimers form transiently during NMD activation. However, our data would argue against the formation of stable homodimers, which mediate the rescue of their respective heterodimerization partner.

Response Fig.: Homodimerization

a, b Western blot after FLAG co-immunoprecipitation (IP) of FLAG-tagged GST (control) or SMG5 (a)/SMG7 (b) constructs in SMG7 KO cells. Clover-tagged SMG5 or SMG7 are co-expressed. Tubulin serves as control.

HEK cells with a SMG7 knockout were generated, which proliferated a bit slower than the parental cells but showed no decrease in viability (Fig. 1). While WT SMG7 and a 14-3-3 mutant could rescue the NMD deficiency, the G100E mutant could not (Fig. 2), suggesting that SMG7 needs to interact with SMG5 for NMD activity. Interestingly, siRNA-mediated knockdowns of SMG5, SMG6 or both together in the SMG7 KO cells led to a much stronger, apparently complete NMD inhibition, indicating a synergistic function between SMG7 and SMG5 and between SMG7 and SMG6. Interestingly, RNA-seq of these conditions with complete NMD inactivation revealed that approx. 40% of the expressed genome of the HEK cells is under direct or indirect control of NMD (Fig. 3), which is much more than previously estimated based on incomplete NMD inhibition. Following up on the indication that SMG6-mediated endonucleolytic cleavage might require SMG5 and SMG7, TPI reporters with an XRN1-resistant pseudoknot structure were used and indeed, when the SMG7 KO was combined with SMG5 KD, no endonucleolytic cleavage was observed anymore. Interestingly, SMG7 KO alone or SMG5 KD alone did not significantly inhibit the endonucleolytic cleavage. The authors then went on to test additional SMG7 mutants for their ability to rescue NMD in the SMG7 KO cells, combined or not with SMG5 or SMG6 KDs. Here is where the manuscript loses focus.

- We thank Oliver Mühlemann for this helpful comment and agree that the order, presentation and focus of results in the previous version of the manuscript could be improved. Motivated by this comment, we re-structured the results part in order to better visualize the detailed functional analysis of SMG5 and SMG7 in two consecutive and conceptually related Figures (now Figs. 2-3), which are in our view easier to understand and accessible to the reader.

This reviewer finds that the reported data raises more questions than it answers and hence suggests to removing that data and instead use it as the starting point for a follow-up manuscript that focuses on the different mechanistic possibilities to decay NMD-sensitive mRNAs. The rescue experiments need to be complemented with other approaches, including IPs in order to become fully informative.

- We thoroughly considered this valid comment and ultimately decided to reorganize the results instead of removing them completely. In combination with novel data e.g. IPs that were added in the revised manuscript, we also feel that the results (even if negative in some cases) are relevant for future studies and will avoid unnecessary experiments by other research groups. Furthermore, we have found compelling contradicting evidence against some rather long-standing views in the field (e.g. the importance of SMG7-C-terminus) that we believe needed to be properly addressed.

Finally, and intriguingly, the authors also found that overexpression of SMG5 in the SMG7 KO cells can rescue the diminished NMD activity, indicating once more that the previously reported SMG5-SMG7 heterodimer formation is not required for NMD. As for Fig. 5, I do not find the rescue experiments with various SMG5 mutants very informative in the absence of complementing approaches and more work would be needed to generate compelling new insights. Among several open questions, it should for example be tested whether the capability of SMG5 to sustain NMD in the absence of SMG7 might require PNRC2.

- We agree that our previous functional analysis of SMG5 was not complete and important aspects were missing. To elucidate those in more detail, we have included new experiments:
 1. Testing if SMG5 activity depends on PNRC2 (Fig. 3c).
 2. Studying the interaction of SMG5 mutants with p-UPF1 in WT or SMG7 KO cells (Fig.3d and Extended Data Fig. 4d).

3. Analyzing SMG5 and SMG5 mutant interactomes by label-free mass spectrometry (Fig. 3e-g).

Based on our results we conclude that SMG5 does not rely on PNRC2 for mediating NMD activity. Furthermore, we show that the expression of a SMG5 mutant lacking the catalytically inactive PIN domain (SMG5 1-853) leads to hyper-phosphorylated UPF1 in WT cells, suggesting a role of the PIN domain in UPF1 dephosphorylation. However, we could not find evidence for stable interactions between SMG5 and protein phosphatases (PP1 or PP2) in a PIN domain-dependent manner. Intriguingly, we found an increased interaction with UPF1 and UPF2 in the mass spectrometry data, suggesting that SMG5 1-853 expression leads to stalled NMD complexes. In summary, our new results allow us to draw a more complete picture about the function of SMG5 in NMD, although further research is needed to address the role of the C-terminus in more detail.

Again, I suggest to spare such additional investigations for a follow-up manuscript instead of showing it here, as the data in its current form rather distracts and confuses the reader.

The Discussion is rather long and attempts to unite the existing NMD literature and their data presented herein into a coherent NMD working model that they call the “two-factor authentication” model. I think this exercise is well done and resulted in an admittedly quite speculative but nevertheless attractive Fig. 7 illustrating that model. It is eventually an editorial decision how much speculative assumptions should be tolerated in working model figures, but this reviewer likes the model and sees its value in that it provides many ideas for follow-up experiments to test specific assumptions of the model. In that sense, the model serves its purpose.

Main points to address:

- For Fig. 2, immunoprecipitations should be performed to confirm that 14-3-3mut indeed fails to interact with p-UPF1 and that G100E fails to interact with SMG5.

- In the revised manuscript we have included results showing that the SMG7 mutants do not interact with the respective binding partner. The co-immunoprecipitation results for FLAG-tagged SMG7 14-3-3mut (which fails to interact with UPF1 and p-UPF1) and G100E mutant (which fails to interact with SMG5) are shown in Fig. 2c.

- The experiment shown in Fig. 4 lacks the SMG6 KD as a control to show that generation of the observed xrFRAG and 3' fragment depends on SMG6 activity. It is assumed, probably based on experiments of that type in a previous publication, but it should be added here again for completeness. Otherwise the conclusion that the SMG5-SMG7 heterodimer is required for SMG6 activity is not warranted.

- We would like to point the reviewer to Extended Data Fig. 6a, in which the dependence of 3' fragment generation on the presence of SMG6 is shown (compare lanes 5-8 to lanes 13-16).

- Since The authors report evidence that SMG5 and SMG7 can functionally substitute for each other, it should be tested in Fig. 4e-j if SMG5 KD in WT HEK cells also leads to increased UPF1 phosphorylation and accordingly increased IP of the known NMD-targeted transcripts with hyperphosphorylated UPF1.

- We thank the reviewer for this helpful comment, which encouraged us to test the impact of isolated SMG5 KD on the functional properties of UPF1. However, we decided to change also other experimental parameters. Previously, we have used overexpressed FLAG-tagged UPF1 constructs to assess the phosphorylation status and RNA binding. We have repeated the RIP and phosphorylation experiments using a) protein G-coupled IP of endogenous UPF1 and b) a

phospho-specific UPF1 antibody in total cell lysates (Fig. 3h and Extended Data Fig. 4f-i). Consistent with our previous experiment using FLAG-tagged UPF1 and a pan-phospho-S/T antibody, we found also with the specific p-UPF1 (S1127) antibody increased UPF1 phosphorylation in SMG7 KO cells. However, no change in phosphorylation could be observed in the SMG5 KD cells. For the RIP experiments using protein G coupled UPF1 antibodies, we no longer detected increased binding to non-NMD targets in SMG7 KO cells. Although we cannot fully explain this discrepancy of RNA binding between endogenous and overexpressed UPF1, we speculate that the impaired target recognition is especially prominent in NMD-impaired (SMG7 KO) cells when UPF1 is more abundant. Since this phenomenon might only occur in non-physiological conditions and for the sake of clarity, we decided to remove the FLAG-UPF1 RIP results. Nevertheless, in the endogenous UPF1 RIP, we do not see any changes in RNA binding when SMG5 was depleted (at least in those 4 targets we investigated).

- Fig. 5 provides a lot of data, but conclusions are difficult because of lacking controls or complementary experiments. To solidify the conclusions drawn from the NMD rescue capacity of the tested SMG7 mutants, IPs should be done to validate that e.g. the 1-633 mutant indeed fails to interact with POP2. Additionally, testing these IPs for co-precipitation of PP2A might be informative.

- To address these important points, we performed a series of immunoprecipitations. However, we were unable to detect the interaction of SMG7 with POP2 or PP2A using either antibodies against endogenous POP2 and PP2A, or using transiently expressed V5-tagged POP2. Considering also the SMG5 mass spectrometry data, we currently do not have evidence that either SMG5 or SMG7 interact with PP2A. We also conclude that under our chosen conditions, the loss of the SMG7-POP2 interaction cannot be tested. However, we have previously established that tethering of the SMG7 C-terminus to reporter mRNAs results in RNA degradation that does not involve endonucleolytic cleavage or 5'-3' decay (PMID: 27917860), which would support the CCR4-NOT-recruitment theory.

Moreover, the finding that in the SMG7 KO / SMG6 KD condition, expression of the truncated SMG7 1-633 construct can rescue NMD begs the question how RNA degradation is triggered in this case. Is the residual SMG6 still sufficient to trigger endonucleolysis or is another pathway taking over, e.g. deadenylation-independent decapping?

- We understand the reviewer's comment, but would like to point out that the SMG7 (1-633) mutant "only" rescues the loss of SMG7 and does not lead to normal NMD levels (comparing log2FC of lane 1 of Luc control KD and lanes 1, 7 of SMG6 KD in Extended Data Fig. 3c). To better visualize the quantitative effects, we now present these qPCR data as heatmaps (Fig. 2f), which allows direct comparison of all conditions. Based on this, we believe that residual SMG6 is responsible for the remaining NMD activity and the SMG7 (1-633) construct displays the same rescue efficiency as full length SMG7.

Collectively, I think that these experiments raise more questions than they answer and would need additional investigation. Rather than including them in this anyway already data-heavy manuscript, they might serve instead as the starting point for another manuscript focusing on the different mechanisms to degrade NMD-sensitive mRNAs.

- We appreciate this comment. However, considering the comments of the other reviewers, we decided to keep these data in the manuscript. We hope that by adding new results and extensively restructuring the results section (including a more focused presentation of

SMG5-SMG7 characterization), we present the results as a well-rounded story, despite some remaining open questions.

- The comment made for Fig. 5 is also valid for Fig. 6: showing that SMG5 can substitute for SMG7 in NMD is compelling, but the rescue experiments need to be complemented with additional experiments in order to be conclusive and hence they should be spared for a follow-up manuscript. Panel 6c (northern blot with Nop65) could be moved to Fig. 4 and replace the current panel 4c there.

- As discussed above, we believe that the revised and restructured SMG5 analyses, including new results, are now more compelling and provide new insights into the mechanism of NMD. Nonetheless, we agree that the NOP56 northern blots could be merged and shown only once (now Fig. 5c,d).

Minor points:

Line 69: The first NMD factors were discovered about 30 years ago, thus “several decades” might be more appropriate than “many decades”.

- This is a valid comment, we have changed the text accordingly.

Fig. 1b and Ext Fig. 1: please indicated what the bands are that are depicted by a *

- Thanks for the remark, the meaning of the * is now explained.

(This report is from Oliver Mühlemann)

Reviewer #3 (Remarks to the Author):

Nonsense-mediated mRNA decay relies on “two-factor authentication” by SMG5-SMG7

Volker Boehm et al.

In this recent work, Gehring and colleagues investigate the role of SMG7 on targeting mRNA to nonsense-mediated RNA decay. Cumulative data from a number of labs have implicated the SMG5-SMG7 heterodimer in binding the core NMD factor, UPF1, and recruiting the CCR4/NOT deadenylation complex to NMD substrates to promote deadenylation and exonucleolytic decay of mRNA targets.

CRISPR knock-out 293T clones lacking detectable SMG7 were generated; these cells demonstrated slower proliferation and increased levels of two characterized NMD transcripts (as monitored by endpoint RT-PCR). RNA-Seq analysis of these clones identified global gene expression changes with NMD-sensitive mRNA isoforms constituting a significant fraction of up-regulated transcripts, as expected. Complementation assays were used to demonstrate that both wild-type and a SMG7 mutant lacking the ability to interact with UPF1 rescue the NMD defect in SMG7 KO cells, whereas a SMG7 mutant unable to interact with SMG5 did not. In experiments in which additional NMD components were knocked down in SMG7 KO cells, depletion of the SMG6 endonuclease was shown to completely abolish NMD, as anticipated when both exonucleolytic and endonucleolytic pathways for targeting NMD substrates to decay are eliminated. Surprisingly, KD of SMG5 in SMG7 KO cells also completely abolished NMD, suggesting that the two decay pathways are not independent as previously assumed. Consistent with this observation, SMG6-mediated endonucleolytic cleavage of both an NMD reporter and the endogenous NMD target, NOP54, were disrupted in cells lacking SMG5-SMG7. Interestingly, while the level of phospho-UPF1 increases in SMG7 KO cells and UPF1 is found to show increased interaction with non-target mRNAs, these increases were not further increased with SMG5 KD (where the largest impact on NMD is observed) and thus are unlikely to underlie NMD inhibition in SMG7/SMG5 KO/KD cells. Moreover, SMG7 deletion mutants unable to interact with the CCR4/NOT deadenylase were able to restore NMD in SMG7 KO, SMG7/SMG5 and SMG7/SMG6 KO/KD cells, indicating a novel role for SMG7 in NMD. Finally, using SMG7 KO/SMG5 KD cells, the authors show that SMG5 over-expression can compensate for loss of SMG7 and that both its 14-3-3 domain (involved in binding UPF1) and the C-terminus (thought to be promote dephosphorylation of UPF1) are required for SMG5 activity in NMD (including SMG6-mediated substrate cleavage).

The data provide novel and convincing evidence for the dependence of SMG5/7 in SMG6-mediated endonucleolytic decay of NMD substrates and questions the role for recruitment of the CCR4/NOT deadenylase in degrading targets. Moreover, the observation that SMG7 deficient in CCR4/NOT binding can restore NMD in cells lacking SMG6 suggests a novel, yet undetermined role for SMG5/7 in NMD. The work falls significantly short in an effort to understand how SMG5/7 are required for SMG6 function (likely through mediating the interaction between UPF1 and SMG6, but untested) or the curious new role for the SMG5/7 proteins in NMD, and instead propose an over-arching model for how these three factors function in NMD based primarily on past literature.

- The reviewer has a fair point in noticing that we do not finally explain the reason why SMG5 and SMG7 are required for SMG6 activity. We agree that the most straight-forward hypothesis would be that SMG5 and SMG7 are needed to allow the access of SMG6 to phosphorylated UPF1. As discussed above with the comment of reviewer #1, we have intensively investigated the dependence of UPF1-SMG6 interaction on SMG5 and/or SMG7.

Using proximity labelling via TurboID-UPF1 we find that many NMD factors are stronger biotinylated in the absence of SMG5-SMG7. We interpret this as one or more different NMD complexes accumulating longer than usual on NMD substrates. Using these data, we developed three alternative hypothetical models that put our new findings in the context of the degradative steps of NMD (see discussion and Fig. 7).

1. The current study provides a multitude of novel and interesting findings from transcriptome analyses, complementation tests and reporter assays; however, the data is generally skimmed over by the authors and it is left to the reader to scrutinize the figures to find the nuggets of supporting data. While this is difficult in itself due to the presentation of the large datasets (and a vague description on how to interpret; i.e. Fig 1f and 3h), some of the most intriguing findings seem not to be fully highlighted - for example, in Fig 5b, the ability of a SMG7 mutant deficient in recruiting the CCR4/NOT deadenylase to complement the SMG7/SMG6 KO/KD cells provides clear evidence for a novel role of SMG7 in NMD outside of deadenylation or an ability to recruit/activate SMG6. Additionally, although data figures show replicates, averages and standard deviations, the text describing the data is highly qualitative.

- This reviewer felt that we did not present our results comprehensively enough. Motivated by this and the reviewer's other comments we aimed to increase the focus of the results section. To this end, we restructured the results section, which should provide a clearer picture of the results. We also revised the discussion to allow sufficient space for discussion of our own data.

2. Strong evidence is presented showing a requirement for SMG5/7 in SMG6-mediated NMD substrate cleavage. One obvious mechanism to explain this observation is through recruitment of SMG6 to UPF1 via SMG5/7, and this should be tested by monitoring SMG6-UPF1 interaction in the various cell lines.

- We have discussed this important point above and in the response to reviewer #1 in more detail and believe that we can provide compelling evidence that SMG6 is still able to interact with UPF1 despite the loss of SMG5 and SMG7. The observation that this interaction is even more pronounced in SMG5-SMG7-depleted cells, rather points to an activating/recycling function of SMG5 and/or SMG7.

3. The authors observe “a complete inhibition of NMD” upon SMG5 or SMG6 KD in SMG7 KO cells. While inhibition of NMD is clearly more pronounced in these cells than in SMG7 knockout alone (Fig 2d), a UPF1 KD control (the ‘gold standard’ in NMD) is needed to help gauge the extent of NMD inhibition in cells depleted of SMG7, SMG5/7 and SMG6/7.

- The reviewer raises an important point here, as UPF1 is the central component of the NMD machinery. It is therefore logical to use a UPF1 KD as gold standard for NMD inhibition. However, we frequently observe in the lab that a “simple” UPF1 KD does not exhibit the same extent of NMD inhibition as we achieve with SMG5/7 or SMG6/7 KDs. To obtain more data to prove this point, we have downloaded and analyzed several datasets from the SRA repository, all of which used UPF1 KDs to inhibit NMD (see Figure below). Although most datasets showed good knockdown of UPF1 (indicated with UPF1 DGE log₂FC) and increased abundance of PTC-containing isoforms, only the effect upon UPF1 KD in the cytoplasmic fraction of HeLa cells showed quantitatively comparable results to our SMG5-SMG7-depleted data (compare panel d and f; UpSet plot in h). Therefore, we believe that our statements made in the manuscript are not unreasonable as the loss of SMG5-SMG7 results in more

efficient NMD inhibition compared to standard UPF1 KD. Nevertheless, we have rephrased our statements, since we cannot rule out that minimal NMD activity still takes place and future improvements might achieve even more exhaustive NMD inhibition.

Boehm, Kueckelmann et al. 2021: Response UPF1 KD

Response Fig.: UPF1 KD

a-f, Volcano plots showing the differential transcript usage (via IsoformSwitchAnalyzeR) in various RNA-Seq data. Isoforms containing GENCODE (release 33) annotated PTC (red, TRUE), regular stop codons (blue, FALSE) or having no annotated open reading frame (gray, NA) are indicated. The change in isoform fraction (dIF) is plotted against the $-\log_{10}$ adjusted p-value (adj.p-value). Density plots show the distribution of filtered isoforms in respect to the dIF, cutoffs were $|dIF| > 0.1$ and adj.p-value < 0.05 . Density plots show the distribution of filtered isoforms in respect to the dIF, cutoffs were $|dIF| > 0.1$ and adj.p-value < 0.05 .

g, Overview of analyzed RNA-seq samples

h, Overlap of upregulated premature termination codon (PTC)-containing isoforms between the RNA-Seq data is shown as UpSet plot.

4. Mutant alleles of SMG7 are expressed at different levels, and so a quantification of IP versus input should be provided for the coIP experiment (Fig 5c).

- According to the reviewer's comment, we now provide quantifications for the SMG7 co-IP experiment shown in Extended Data Fig. 3f.

4. Ectopic expression of wild-type SMG5 or the G120E mutant (unable to bind SMG7) is shown to complement SMG7 KO (Fig 6b). It needs to be clarified that SMG5 is over-expressed in these cells and some indication of its level over endogenous SMG5 (and compare to endogenous SMG7) provided. It is reasonable to assume that SMG5 can form homodimers in this context that function in NMD similarly to SMG5/7 heterodimers, however, this is not tested or discussed.

- We have quantified the level of overexpression (Extended Data Fig. 4c) and state this now explicitly in the text. We have also tested whether SMG5 and SMG7 can form homodimers when overexpressed (see comment to reviewer #1). However, we did not find any evidence for this.

6. The authors convincingly show in Fig 1 that their SMG7 KO clones have stronger NMD defects than a SMG7 KO (either in their hands or based on previous data by Colombo et al using HeLa cells). This is not unexpected. Moreover, the detailed comparison of transcriptome changes between the SMG7 KO in 293T cells and SMG7 KD in HeLa is comparing apples-to-oranges and distracts from other key findings in the paper.

- We agree with the reviewer on the point that we should not “over-discuss” the differences between SMG7 KD in HeLa cells and the SMG7 KO in 293 cells. However, we believe that the initial analyses (Fig.1e-i and Extended Data Fig. 2b-e) are required to establish that the KO of SMG7 is indeed more effective than the KD. Nevertheless, we have removed superfluous analyses and the corresponding text (e.g. barcode plots).

7. The discussion section of the manuscript and model figure presented in Fig 7 provides a comprehensive overview of NMD events based primarily on published literature and is better suited for a review article. Critically, the authors fail to hone-in on the novel findings gained from their study and how these observations should change how we think about events occurring between (presumably) the NMD machinery and terminating ribosome. There are many provocative findings from this study and the impact of their data and novel roles for SMG5 and SMG7 in eliciting NMD should be the focus of the discussion and better reflected in their model figure(s) and in the manuscript title.

- We completely agree with the reviewer's statement that we should have focused more on our novel findings in the model and discussion section of the previous version of the manuscript. Since we received also concordant feedback from the other reviewers, we completely exchanged our model to better reflect the important implications of our work. Specifically, we show and discuss now three mechanistic hypotheses for the role of SMG5-SMG7 in NMD in more detail.

REVIEWERS' COMMENTS

Reviewer #1 (Remarks to the Author):

The authors have made substantial changes to the revised manuscript, which has resulted in a robust, clear report. The Turbo-ID study and the modified discussion with three possible hypotheses of how SMG5-7 could act upstream of SMG6 are a great addition. This study will hopefully trigger further research on the obvious open questions of how SMG6 is activated or authenticated and how NMD progresses in cells.

In summary, I have no further questions and support the publication of the manuscript.

Reviewer #2 (Remarks to the Author):

The authors have thoroughly revised their initial version of the manuscript based on the reviewers' comments and the revised version is now much more focused on the main findings, better structured and hence easier and more pleasant to read. Also the revised illustration of the "two factor authentication model" (Fig. 7) has been improved and become clearer. I congratulate the authors to this overall high quality and interesting piece of work, which provides exciting new insights into the still enigmatic mechanism of NMD activation.

My points have all been satisfactorily addressed and I recommend publication of the manuscript in its present form.

(This review is from Oliver Mühlemann)

Reviewer #3 (Remarks to the Author):

Nonsense-mediated mRNA decay relies on "two-factor authentication" by SMG5-SMG7

In their revised manuscript, Boehm et al. address the majority of the concerns raised by this and the other reviewers. The inclusion of important controls, extended analyses, and co-immunoprecipitation and proximity labeling data all strengthen the conclusions and help shed light on the requirement of SMG5-SMG7 for SMG6 activity on NMD substrates. Additionally, they have done a worthy job at rewriting the results and discussion to better highlight the key findings from their data. The manuscript is now much clearer and - with the additional supporting data - provides convincing evidence for the interdependence of SMG5-SMG7 and SMG6 in targeting substrates to NMD.

The only remaining comments relate to both the manuscript title and models they offer to represent their findings. While there is indeed evidence of a hierarchical requirement for NMD factors (and likely significant mRNP remodeling) in targeting of NMD substrates, the concept of two 'authentication' steps is not as clear and perhaps misleading to readers, given the lack of full understanding of the events. Likewise, although provocative, evoking this authentication concept in the title does little to describe the major (and important) mechanistic findings presented in the paper. A title such as "Endonucleolytic cleavage of NMD targets by SMG6 relies on SMG5-SMG7" is much more informative to the reader. Finally, presentation of three possible models to explain their findings (Fig. 7) is confusing, given that at least one of these (7a) is not supported by their own data. Presentation of one favored model would be preferred.

General response to the reviewers

We very much appreciate the interest of all three reviewers in our work and are grateful for their thorough evaluation of the manuscript and for their constructive criticisms. In the revised version we have addressed most of their comments by modifying the text and by including new experimental data as described in this detailed point-by-point response. Text changes made to the manuscript during the revision are tracked with Word's 'Track Changes'.

The highlights of the revised manuscript include:

- Restructured results part with condensed functional characterization of SMG5 and SMG7 to improve readability.
- Various additional aspects addressed, e.g. the role of PNRC2 for SMG5 activity.
- New interaction data using SMG5-SMG7 co-IPs, UPF1 Turbo-ID proximity labelling and mass spectrometry supporting the “two-factor authentication” model.
- Substantially revised discussion and model.

REVIEWER COMMENTS – first round of revision

Reviewer #1 (Remarks to the Author):

The manuscript entitled “Nonsense-mediated mRNA decay relies on “two-factor authentication by SMG5-SMG7” by Boehm, Kueckelmann et al. addresses the mechanism of target recognition in the NMD pathway by the SMG5, SMG6 and SMG7 proteins, together with phosphorylated UPF1. The prevailing model of NMD in metazoans suggests that the resident time of UPF1 on an mRNA transcript is an indicator of NMD and that hyperphosphorylation of UPF1 and its subsequent recognition by SMG5, SMG7 and SMG6 commits a UPF1-bound mRNA to decay through NMD. The proteins SMG5 and SMG7 form a heterodimer, which was earlier shown to bridge the NMD machinery to the CCR4-NOT deadenylation complex. SMG6, on the other hand, possesses endonucleolytic activity and plays an important role in NMD. These proteins were shown to interact with phosphorylated UPF1 through their TPR domains, although SMG6 was also shown to interact with UPF1 in a phosphorylation-independent manner. Based on knockdown and rescue studies, the roles of SMG5-7 and SMG6 were thought to be interchangeable, owing to a functional redundancy among the proteins.

In this study, the authors generated a SMG7 KO cell-line and used this tool to investigate the mechanism of action of SMG7, SMG5 and SMG6 in further detail. They found, using RT-PCR assays on specific targets as well as RNA-Seq, that deletion of SMG7 significantly abolished NMD and leads to upregulation of several transcripts. This is in contrast to previous studies with siRNA-mediated SMG7 knockdown where the effects were much milder. At that time, this was presumed to be due to compensatory effect of SMG6. Interestingly siRNA KD of SMG5 in the background of SMG7 KO further heightened the effect on NMD. The effect on NMD could be rescued by addition of wildtype SMG7 and a SMG7 mutant that is incapable of binding phospho-UPF1, but not one that is incapable of binding SMG5. Furthermore, a SMG7 construct lacking the C-terminus (which was proposed to recruit the deadenylation machinery) is capable of rescuing NMD. These new data cast a doubt on the hitherto proposed function of SMG7.

SMG5 was largely capable of rescuing the effect of SMG7 KO, though surprisingly the catalytically inactive PIN domain was found to be essential to recapitulate this effect. Depletion of SMG5 together with a complete KO of SMG7 also affected the endonucleolytic activity of SMG6.

The data presented in this paper are convincing and presented very clearly. The Methods section is sufficiently detailed. The extended data included also support the figures in the main text.

- We thank the reviewer for the overall positive feedback on our manuscript.

Major Points:

- It would be very helpful if the authors could indicate the extent of knockdown in each of the siRNA treated samples (in terms of residual protein or RNA) as it gives the reader a feeling for how strong the effects of knockdown of SMG5 and SMG6 are.

- This is a good suggestion because the strength of the knockdown is likely to be critical for the strength of the expected effects. We have included exemplary westerns blots as **Extended Data Fig. 3a**, showing the residual amount of SMG5, SMG6 and SMG7 proteins after the respective knockdowns. A dilution series of the control sample was included to allow the reader to estimate the KD efficiencies. Furthermore, **Supplementary Table 1** provides information on the RNA level about the knockdown of SMG5 and SMG6 in the RNA-seq data. A differential gene expression analysis of all NMD factors in the RNA-seq data is now also displayed in a comprehensive heatmap in **Extended Data Fig. 5a**.

- From the data presented here, it is clear that there is a functional dependency of SMG6 on SMG5 and SMG7. However, the hierarchy in terms of action of the proteins is not convincing. It appears that at any given point of time in the cell, a combination of two of the three proteins (SMG5/6/7) is essential to mediate NMD. It is possible that the main scaffolding protein is SMG7, which is why knockout of SMG7 has a strong effect. It must be noted (as shown in Figure 2D) that KD of SMG6 in SMG7-KO cells also has a fairly strong effect, similar to that of SMG5. Although the TPR domain of SMG6 does not interact with SMG5 or SMG7, it would be worthwhile to test if full-length SMG6 interacts with either SMG7 or SMG5, and if this interaction is necessary for stability or function of SMG6. As such, from this manuscript it is not clear why SMG5-7 should be necessary for SMG6 activity.

- We agree that the reviewer's hypothesis that "two out of the three SMG5/6/7 proteins might be enough to mediate NMD" is very suggestive. However, we did not see any additive effects of the combined SMG5 + SMG6 KD over isolated SMG6 depletion (**Fig. 2e** and **Extended Data Fig. 3b**), which one would expect if the hypothesis was true.
- To test if indeed SMG5 or SMG7 could interact with full-length SMG6, we performed several experiments. Standard FLAG-IPs with FLAG-tagged SMG7 or SMG5 did not show any detectable interaction with endogenous SMG6. Furthermore, we have performed label-free mass spectrometry analyses of the SMG5 interactome and could not detect any co-immunoprecipitated SMG6 (**Supplementary Table 4**). Furthermore, our new UPF1 TurboID proximity labelling data now shed light on the function of SMG5-SMG7 for NMD in general and the activity of SMG6 in particular (**Fig.6**, **Extended Data Fig. 7** and **Supplementary Table 5**). In the absence of SMG5-SMG7, we find that many NMD factors (including SMG6) are stronger biotinylated by TurboID-UPF1, which - in combination with the functional data - corresponds in our view to the formation of inactive NMD complexes. Using these data, we developed several models that put our new findings in the context of NMD (see discussion and **Fig. 7**).

- The authors speculate that binding of SMG5-7 to phosphorylated UPF1 induces a conformational change within UPF1, which might facilitate binding of SMG6. However, there is no evidence to support this statement, which is an important step in the "two-step authentication model". This

would require careful analysis of binding of SMG6 to UPF1 in the presence and absence of SMG5-7 and upon different phosphorylation of UPF1 to different extents, which might be out of the scope of this manuscript.

- We agree that crucial evidence supporting our speculation about SMG5-SMG7 potentially inducing a conformational change within UPF1 was missing from the previous version of the manuscript. Therefore, we have now removed most of this speculative aspect from the discussion as part of the revision.
 - However, the reviewers' questions about SMG6-UPF1 interaction prompted us to conduct further experiments. We have intensively tried to obtain reliable data showing whether any changes in the interaction between UPF1 and SMG6 occur in response to the presence or absence of SMG5 and/or SMG7. Standard IPs with either FLAG-tagged SMG6 or UPF1 were unsuccessful and did not consistently showed convincing binding to the other potential interaction partner over background controls. We have therefore switched to using the TurboID-mediated proximity labeling with the aim to capture transient interactions. To this end, we tried to express TurboID-tagged SMG6 and to check for transient interactions by proximity labeling, which failed due to the poor expression and therefore low labeling efficiency of this SMG6 construct. Conversely, TurboID-tagged UPF1 performed well, displayed convincing biotinylation of known interaction partners (e.g. UPF3B or STAU2). As outlined above, using TurboID-UPF1 we find that many NMD factors, including those responsible for the first authentication step, are stronger "proximity-labelled" in the absence of SMG5-SMG7. We suggest that this corresponds to the formation of SMG6-containing, inactive NMD complexes, which are at least temporarily arrested on NMD substrates. We feel that with this experiment and the resulting changes in the results section and discussion, we have now answered the reviewer's - admittedly legitimate - question.
- Furthermore, establishing a hierarchy in the NMD pathway calls for demonstrating that depletion of SMG6 does not impact the activity of SMG5-7 at all. KD of SMG6 does in fact show a slight increase of the target transcripts, as shown in Figure 2D.
 - We agree with the reviewer that the KD of SMG6 stabilizes many NMD targets, which is consistent with the hypothesis that SMG6-mediated endonucleolytic cleavage is the major degradation pathway of NMD. Moreover, SMG6 activity can be monitored by visualizing the degradation intermediates of the endonucleolytic cleavage (3' fragments). However, the degradative activity of SMG5-SMG7 is more difficult to study. While SMG7 has been reported to lead to accelerated deadenylation via the recruitment of the CCR4-NOT complex, we found that the C-terminal deletion mutant (which supposedly fails to interact with CNOT8/POP2, a component of the CCR4-NOT complex) fully rescued SMG7 KO. It is also highly peculiar that SMG5 - with no reported degradative ability - was able to rescue the SMG7 KO. As motivated by the comment of Oliver Mühlemann (see below), we also established that SMG5 does not rely on PNRC2 for NMD activity (Fig. 3c). Therefore, it is currently unclear how to properly measure the activity of SMG5 or SMG7 and further research is needed to resolve this issue.
 - Nevertheless, we understand the reviewer's objections to defining a strict hierarchy that would preclude the possibility that SMG6 also influences SMG5-7 activities. This aspect was not clearly discussed previously in the manuscript and we have now addressed this issue in the revised discussion. In summary, our data do not allow us to rule out the possibility that SMG6 has an effect on the activity of SMG5 or SMG7.

A minor point in addition: the last sentence in the figure legend of 1g-i is repeated (Density plots show the distribution...)

- This error was corrected.

In summary, I suggest revising the discussion of the manuscript to focus only on the functional redundancy of SMG5, SMG7 and SMG6, which on its own is fairly interesting (albeit to NMD aficionados).

- We took this concluding remark very seriously and restructured major parts of the manuscript and, in particular, rewrote the discussion. With these major revisions we wanted to increase the comprehensibility of the manuscript, to establish a more logical order of the experiments, and to focus the discussion on the most important findings of the work.

Reviewer #2 (Remarks to the Author):

This study from the Gehring lab provides convincing evidence against the currently popular idea in the NMD field positing that in mammalian cells, degradation of NMD-targeted mRNAs can be initiated independently by either of two pathways, namely by SMG6-mediated endonucleolytic cleavage near the stop codon or by SMG5-SMG7-mediated recruitment of the CCR4-NOT deadenylase complex. Boehm and colleagues now show that the combined loss of SMG5 and SMG7 completely inactivates NMD. Their findings are consistent with another recently published preprint by the Leeb lab (Galimberti et al., bioRxiv 10.1101/2020.07.07.180133v1), so there is no doubt anymore that SMG6 activity requires the presence of at least one of the two heterodimerizing factors, SMG5 or SMG7.

Based on this data, the authors propose a “two-factor authentication” model to activate SMG6-mediated mRNA cleavage, in which hyperphosphorylation represents the first factor and SMG5-SMG7 recruitment the second factor. Moreover, the authors report the surprising finding that SMG5 can functionally substitute for SMG7 and vice versa, but it remains unclear what exact function in the NMD pathway both of these proteins are able to execute in the absence of the other. It would be highly insightful to check if SMG5 and SMG7 form homodimers if the normal binding partner is absent.

- We thank Oliver Mühlemann for this suggestion. For us, the formation of SMG5-SMG5 or SMG7-SMG7 homodimers also seemed a logical explanation for the observed rescue effects in SMG5-SMG7. Accordingly, we investigated homodimerization of both SMG5 and SMG7 by immunoprecipitation from cell lysates, expressing simultaneously both FLAG- and clover-tagged versions of either SMG5 or SMG7. We used comparable conditions to those that were used for the rescue assays, because we wanted to avoid studying homodimerization *in vitro*, which may not occur in living cells. We used GST as a background control and the interaction between the “true” partner SMG5 or SMG7 as a positive control. In addition, we also used the interaction-defective mutants of SMG5 and SMG7, respectively, to learn more about the molecular details of the potential homodimers. However, neither SMG5 nor SMG7 appeared to form homodimers that could be detected with our experimental approach (see Figure below). We cannot rule out the possibility that homodimers form transiently during NMD activation. However, our data would argue against the formation of stable homodimers, which mediate the rescue of their respective heterodimerization partner.

Response Fig.: Homodimerization

a, b Western blot after FLAG co-immunoprecipitation (IP) of FLAG-tagged GST (control) or SMG5 (a)/SMG7 (b) constructs in SMG7 KO cells. Clover-tagged SMG5 or SMG7 are co-expressed. Tubulin serves as control.

HEK cells with a SMG7 knockout were generated, which proliferated a bit slower than the parental cells but showed no decrease in viability (Fig. 1). While WT SMG7 and a 14-3-3 mutant could rescue the NMD deficiency, the G100E mutant could not (Fig. 2), suggesting that SMG7 needs to interact with SMG5 for NMD activity. Interestingly, siRNA-mediated knockdowns of SMG5, SMG6 or both together in the SMG7 KO cells led to a much stronger, apparently complete NMD inhibition, indicating a synergistic function between SMG7 and SMG5 and between SMG7 and SMG6. Interestingly, RNA-seq of these conditions with complete NMD inactivation revealed that approx. 40% of the expressed genome of the HEK cells is under direct or indirect control of NMD (Fig. 3), which is much more than previously estimated based on incomplete NMD inhibition. Following up on the indication that SMG6-mediated endonucleolytic cleavage might require SMG5 and SMG7, TPI reporters with an XRN1-resistant pseudoknot structure were used and indeed, when the SMG7 KO was combined with SMG5 KD, no endonucleolytic cleavage was observed anymore. Interestingly, SMG7 KO alone or SMG5 KD alone did not significantly inhibit the endonucleolytic cleavage. The authors then went on to test additional SMG7 mutants for their ability to rescue NMD in the SMG7 KO cells, combined or not with SMG5 or SMG6 KDs. Here is where the manuscript loses focus.

- We thank Oliver Mühlemann for this helpful comment and agree that the order, presentation and focus of results in the previous version of the manuscript could be improved. Motivated by this comment, we re-structured the results part in order to better visualize the detailed functional analysis of SMG5 and SMG7 in two consecutive and conceptually related Figures (now Figs. 2-3), which are in our view easier to understand and accessible to the reader.

This reviewer finds that the reported data raises more questions than it answers and hence suggests to removing that data and instead use it as the starting point for a follow-up manuscript that focuses on the different mechanistic possibilities to decay NMD-sensitive mRNAs. The rescue experiments need to be complemented with other approaches, including IPs in order to become fully informative.

- We thoroughly considered this valid comment and ultimately decided to reorganize the results instead of removing them completely. In combination with novel data e.g. IPs that were added in the revised manuscript, we also feel that the results (even if negative in some cases) are relevant for future studies and will avoid unnecessary experiments by other research groups. Furthermore, we have found compelling contradicting evidence against some rather long-standing views in the field (e.g. the importance of SMG7-C-terminus) that we believe needed to be properly addressed.

Finally, and intriguingly, the authors also found that overexpression of SMG5 in the SMG7 KO cells can rescue the diminished NMD activity, indicating once more that the previously reported SMG5-SMG7 heterodimer formation is not required for NMD. As for Fig. 5, I do not find the rescue experiments with various SMG5 mutants very informative in the absence of complementing approaches and more work would be needed to generate compelling new insights. Among several open questions, it should for example be tested whether the capability of SMG5 to sustain NMD in the absence of SMG7 might require PNRC2.

- We agree that our previous functional analysis of SMG5 was not complete and important aspects were missing. To elucidate those in more detail, we have included new experiments:
 1. Testing if SMG5 activity depends on PNRC2 (Fig. 3c).
 2. Studying the interaction of SMG5 mutants with p-UPF1 in WT or SMG7 KO cells (Fig.3d and Extended Data Fig. 4d).

3. Analyzing SMG5 and SMG5 mutant interactomes by label-free mass spectrometry (Fig. 3e-g).

Based on our results we conclude that SMG5 does not rely on PNRC2 for mediating NMD activity. Furthermore, we show that the expression of a SMG5 mutant lacking the catalytically inactive PIN domain (SMG5 1-853) leads to hyper-phosphorylated UPF1 in WT cells, suggesting a role of the PIN domain in UPF1 dephosphorylation. However, we could not find evidence for stable interactions between SMG5 and protein phosphatases (PP1 or PP2) in a PIN domain-dependent manner. Intriguingly, we found an increased interaction with UPF1 and UPF2 in the mass spectrometry data, suggesting that SMG5 1-853 expression leads to stalled NMD complexes. In summary, our new results allow us to draw a more complete picture about the function of SMG5 in NMD, although further research is needed to address the role of the C-terminus in more detail.

Again, I suggest to spare such additional investigations for a follow-up manuscript instead of showing it here, as the data in its current form rather distracts and confuses the reader.

The Discussion is rather long and attempts to unite the existing NMD literature and their data presented herein into a coherent NMD working model that they call the “two-factor authentication” model. I think this exercise is well done and resulted in an admittedly quite speculative but nevertheless attractive Fig. 7 illustrating that model. It is eventually an editorial decision how much speculative assumptions should be tolerated in working model figures, but this reviewer likes the model and sees its value in that it provides many ideas for follow-up experiments to test specific assumptions of the model. In that sense, the model serves its purpose.

Main points to address:

- For Fig. 2, immunoprecipitations should be performed to confirm that 14-3-3mut indeed fails to interact with p-UPF1 and that G100E fails to interact with SMG5.

- In the revised manuscript we have included results showing that the SMG7 mutants do not interact with the respective binding partner. The co-immunoprecipitation results for FLAG-tagged SMG7 14-3-3mut (which fails to interact with UPF1 and p-UPF1) and G100E mutant (which fails to interact with SMG5) are shown in Fig. 2c.

- The experiment shown in Fig. 4 lacks the SMG6 KD as a control to show that generation of the observed xrFRAG and 3' fragment depends on SMG6 activity. It is assumed, probably based on experiments of that type in a previous publication, but it should be added here again for completeness. Otherwise the conclusion that the SMG5-SMG7 heterodimer is required for SMG6 activity is not warranted.

- We would like to point the reviewer to Extended Data Fig. 6a, in which the dependence of 3' fragment generation on the presence of SMG6 is shown (compare lanes 5-8 to lanes 13-16).

- Since The authors report evidence that SMG5 and SMG7 can functionally substitute for each other, it should be tested in Fig. 4e-j if SMG5 KD in WT HEK cells also leads to increased UPF1 phosphorylation and accordingly increased IP of the known NMD-targeted transcripts with hyperphosphorylated UPF1.

- We thank the reviewer for this helpful comment, which encouraged us to test the impact of isolated SMG5 KD on the functional properties of UPF1. However, we decided to change also other experimental parameters. Previously, we have used overexpressed FLAG-tagged UPF1 constructs to assess the phosphorylation status and RNA binding. We have repeated the RIP and phosphorylation experiments using a) protein G-coupled IP of endogenous UPF1 and b) a

phospho-specific UPF1 antibody in total cell lysates (Fig. 3h and Extended Data Fig. 4f-i). Consistent with our previous experiment using FLAG-tagged UPF1 and a pan-phospho-S/T antibody, we found also with the specific p-UPF1 (S1127) antibody increased UPF1 phosphorylation in SMG7 KO cells. However, no change in phosphorylation could be observed in the SMG5 KD cells. For the RIP experiments using protein G coupled UPF1 antibodies, we no longer detected increased binding to non-NMD targets in SMG7 KO cells. Although we cannot fully explain this discrepancy of RNA binding between endogenous and overexpressed UPF1, we speculate that the impaired target recognition is especially prominent in NMD-impaired (SMG7 KO) cells when UPF1 is more abundant. Since this phenomenon might only occur in non-physiological conditions and for the sake of clarity, we decided to remove the FLAG-UPF1 RIP results. Nevertheless, in the endogenous UPF1 RIP, we do not see any changes in RNA binding when SMG5 was depleted (at least in those 4 targets we investigated).

- Fig. 5 provides a lot of data, but conclusions are difficult because of lacking controls or complementary experiments. To solidify the conclusions drawn from the NMD rescue capacity of the tested SMG7 mutants, IPs should be done to validate that e.g. the 1-633 mutant indeed fails to interact with POP2. Additionally, testing these IPs for co-precipitation of PP2A might be informative.

- To address these important points, we performed a series of immunoprecipitations. However, we were unable to detect the interaction of SMG7 with POP2 or PP2A using either antibodies against endogenous POP2 and PP2A, or using transiently expressed V5-tagged POP2. Considering also the SMG5 mass spectrometry data, we currently do not have evidence that either SMG5 or SMG7 interact with PP2A. We also conclude that under our chosen conditions, the loss of the SMG7-POP2 interaction cannot be tested. However, we have previously established that tethering of the SMG7 C-terminus to reporter mRNAs results in RNA degradation that does not involve endonucleolytic cleavage or 5'-3' decay (PMID: 27917860), which would support the CCR4-NOT-recruitment theory.

Moreover, the finding that in the SMG7 KO / SMG6 KD condition, expression of the truncated SMG7 1-633 construct can rescue NMD begs the question how RNA degradation is triggered in this case. Is the residual SMG6 still sufficient to trigger endonucleolysis or is another pathway taking over, e.g. deadenylation-independent decapping?

- We understand the reviewer's comment, but would like to point out that the SMG7 (1-633) mutant "only" rescues the loss of SMG7 and does not lead to normal NMD levels (comparing log2FC of lane 1 of Luc control KD and lanes 1, 7 of SMG6 KD in Extended Data Fig. 3c). To better visualize the quantitative effects, we now present these qPCR data as heatmaps (Fig. 2f), which allows direct comparison of all conditions. Based on this, we believe that residual SMG6 is responsible for the remaining NMD activity and the SMG7 (1-633) construct displays the same rescue efficiency as full length SMG7.

Collectively, I think that these experiments raise more questions than they answer and would need additional investigation. Rather than including them in this anyway already data-heavy manuscript, they might serve instead as the starting point for another manuscript focusing on the different mechanisms to degrade NMD-sensitive mRNAs.

- We appreciate this comment. However, considering the comments of the other reviewers, we decided to keep these data in the manuscript. We hope that by adding new results and extensively restructuring the results section (including a more focused presentation of

SMG5-SMG7 characterization), we present the results as a well-rounded story, despite some remaining open questions.

- The comment made for Fig. 5 is also valid for Fig. 6: showing that SMG5 can substitute for SMG7 in NMD is compelling, but the rescue experiments need to be complemented with additional experiments in order to be conclusive and hence they should be spared for a follow-up manuscript. Panel 6c (northern blot with Nop65) could be moved to Fig. 4 and replace the current panel 4c there.

- As discussed above, we believe that the revised and restructured SMG5 analyses, including new results, are now more compelling and provide new insights into the mechanism of NMD. Nonetheless, we agree that the NOP56 northern blots could be merged and shown only once (now Fig. 5c,d).

Minor points:

Line 69: The first NMD factors were discovered about 30 years ago, thus “several decades” might be more appropriate than “many decades”.

- This is a valid comment, we have changed the text accordingly.

Fig. 1b and Ext Fig. 1: please indicated what the bands are that are depicted by a *

- Thanks for the remark, the meaning of the * is now explained.

(This report is from Oliver Mühlemann)

Reviewer #3 (Remarks to the Author):

Nonsense-mediated mRNA decay relies on “two-factor authentication” by SMG5-SMG7

Volker Boehm et al.

In this recent work, Gehring and colleagues investigate the role of SMG7 on targeting mRNA to nonsense-mediated RNA decay. Cumulative data from a number of labs have implicated the SMG5-SMG7 heterodimer in binding the core NMD factor, UPF1, and recruiting the CCR4/NOT deadenylation complex to NMD substrates to promote deadenylation and exonucleolytic decay of mRNA targets.

CRISPR knock-out 293T clones lacking detectable SMG7 were generated; these cells demonstrated slower proliferation and increased levels of two characterized NMD transcripts (as monitored by end-point RT-PCR). RNA-Seq analysis of these clones identified global gene expression changes with NMD-sensitive mRNA isoforms constituting a significant fraction of up-regulated transcripts, as expected. Complementation assays were used to demonstrate that both wild-type and a SMG7 mutant lacking the ability to interact with UPF1 rescue the NMD defect in SMG7 KO cells, whereas a SMG7 mutant unable to interact with SMG5 did not. In experiments in which additional NMD components were knocked down in SMG7 KO cells, depletion of the SMG6 endonuclease was shown to completely abolish NMD, as anticipated when both exonucleolytic and endonucleolytic pathways for targeting NMD substrates to decay are eliminated. Surprisingly, KD of SMG5 in SMG7 KO cells also completely abolished NMD, suggesting that the two decay pathways are not independent as previously assumed. Consistent with this observation, SMG6-mediated endonucleolytic cleavage of both an NMD reporter and the endogenous NMD target, NOP54, were disrupted in cells lacking SMG5-SMG7. Interestingly, while the level of phospho-UPF1 increases in SMG7 KO cells and UPF1 is found to show increased interaction with non-target mRNAs, these increases were not further increased with SMG5 KD (where the largest impact on NMD is observed) and thus are unlikely to underlie NMD inhibition in SMG7/SMG5 KO/KD cells. Moreover, SMG7 deletion mutants unable to interact with the CCR4/NOT deadenylase were able to restore NMD in SMG7 KO, SMG7/SMG5 and SMG7/SMG6 KO/KD cells, indicating a novel role for SMG7 in NMD. Finally, using SMG7 KO/SMG5 KD cells, the authors show that SMG5 over-expression can compensate for loss of SMG7 and that both its 14-3-3 domain (involved in binding UPF1) and the C-terminus (thought to be promote dephosphorylation of UPF1) are required for SMG5 activity in NMD (including SMG6-mediated substrate cleavage).

The data provide novel and convincing evidence for the dependence of SMG5/7 in SMG6-mediated endonucleolytic decay of NMD substrates and questions the role for recruitment of the CCR4/NOT deadenylase in degrading targets. Moreover, the observation that SMG7 deficient in CCR4/NOT binding can restore NMD in cells lacking SMG6 suggests a novel, yet undetermined role for SMG5/7 in NMD. The work falls significantly short in an effort to understand how SMG5/7 are required for SMG6 function (likely through mediating the interaction between UPF1 and SMG6, but untested) or the curious new role for the SMG5/7 proteins in NMD, and instead propose an over-arching model for how these three factors function in NMD based primarily on past literature.

- The reviewer has a fair point in noticing that we do not finally explain the reason why SMG5 and SMG7 are required for SMG6 activity. We agree that the most straight-forward hypothesis would be that SMG5 and SMG7 are needed to allow the access of SMG6 to phosphorylated UPF1. As discussed above with the comment of reviewer #1, we have intensively investigated the dependence of UPF1-SMG6 interaction on SMG5 and/or SMG7.

Using proximity labelling via TurboID-UPF1 we find that many NMD factors are stronger biotinylated in the absence of SMG5-SMG7. We interpret this as one or more different NMD complexes accumulating longer than usual on NMD substrates. Using these data, we developed three alternative hypothetical models that put our new findings in the context of the degradative steps of NMD (see discussion and Fig. 7).

1. The current study provides a multitude of novel and interesting findings from transcriptome analyses, complementation tests and reporter assays; however, the data is generally skimmed over by the authors and it is left to the reader to scrutinize the figures to find the nuggets of supporting data. While this is difficult in itself due to the presentation of the large datasets (and a vague description on how to interpret; i.e. Fig 1f and 3h), some of the most intriguing findings seem not to be fully highlighted - for example, in Fig 5b, the ability of a SMG7 mutant deficient in recruiting the CCR4/NOT deadenylase to complement the SMG7/SMG6 KO/KD cells provides clear evidence for a novel role of SMG7 in NMD outside of deadenylation or an ability to recruit/activate SMG6. Additionally, although data figures show replicates, averages and standard deviations, the text describing the data is highly qualitative.

- This reviewer felt that we did not present our results comprehensively enough. Motivated by this and the reviewer's other comments we aimed to increase the focus of the results section. To this end, we restructured the results section, which should provide a clearer picture of the results. We also revised the discussion to allow sufficient space for discussion of our own data.

2. Strong evidence is presented showing a requirement for SMG5/7 in SMG6-mediated NMD substrate cleavage. One obvious mechanism to explain this observation is through recruitment of SMG6 to UPF1 via SMG5/7, and this should be tested by monitoring SMG6-UPF1 interaction in the various cell lines.

- We have discussed this important point above and in the response to reviewer #1 in more detail and believe that we can provide compelling evidence that SMG6 is still able to interact with UPF1 despite the loss of SMG5 and SMG7. The observation that this interaction is even more pronounced in SMG5-SMG7-depleted cells, rather points to an activating/recycling function of SMG5 and/or SMG7.

3. The authors observe “a complete inhibition of NMD” upon SMG5 or SMG6 KD in SMG7 KO cells. While inhibition of NMD is clearly more pronounced in these cells than in SMG7 knockout alone (Fig 2d), a UPF1 KD control (the ‘gold standard’ in NMD) is needed to help gauge the extent of NMD inhibition in cells depleted of SMG7, SMG5/7 and SMG6/7.

- The reviewer raises an important point here, as UPF1 is the central component of the NMD machinery. It is therefore logical to use a UPF1 KD as gold standard for NMD inhibition. However, we frequently observe in the lab that a “simple” UPF1 KD does not exhibit the same extent of NMD inhibition as we achieve with SMG5/7 or SMG6/7 KDs. To obtain more data to prove this point, we have downloaded and analyzed several datasets from the SRA repository, all of which used UPF1 KDs to inhibit NMD (see Figure below). Although most datasets showed good knockdown of UPF1 (indicated with UPF1 DGE log2FC) and increased abundance of PTC-containing isoforms, only the effect upon UPF1 KD in the cytoplasmic fraction of HeLa cells showed quantitatively comparable results to our SMG5-SMG7-depleted data (compare panel d and f; UpSet plot in h). Therefore, we believe that our statements made in the manuscript are not unreasonable as the loss of SMG5-SMG7 results in more

efficient NMD inhibition compared to standard UPF1 KD. Nevertheless, we have rephrased our statements, since we cannot rule out that minimal NMD activity still takes place and future improvements might achieve even more exhaustive NMD inhibition.

Boehm, Kueckelmann et al. 2021: Response UPF1 KD

Response Fig.: UPF1 KD

a-f, Volcano plots showing the differential transcript usage (via IsoformSwitchAnalyzeR) in various RNA-Seq data. Isoforms containing GENCODE (release 33) annotated PTC (red, TRUE), regular stop codons (blue, FALSE) or having no annotated open reading frame (gray, NA) are indicated. The change in isoform fraction (dIF) is plotted against the -log10 adjusted p-value (adj.p-value). Density plots show the distribution of filtered isoforms in respect to the dIF, cutoffs were |dIF| > 0.1 and adj.p-value < 0.05. Density plots show the distribution of filtered isoforms in respect to the dIF, cutoffs were |dIF| > 0.1 and adj.p-value < 0.05.

g, Overview of analyzed RNA-seq samples

h, Overlap of upregulated premature termination codon (PTC)-containing isoforms between the RNA-Seq data is shown as UpSet plot.

4. Mutant alleles of SMG7 are expressed at different levels, and so a quantification of IP versus input should be provided for the coIP experiment (Fig 5c).

- According to the reviewer's comment, we now provide quantifications for the SMG7 co-IP experiment shown in Extended Data Fig. 3f.

4. Ectopic expression of wild-type SMG5 or the G120E mutant (unable to bind SMG7) is shown to complement SMG7 KO (Fig 6b). It needs to be clarified that SMG5 is over-expressed in these cells and some indication of its level over endogenous SMG5 (and compare to endogenous SMG7) provided. It is reasonable to assume that SMG5 can form homodimers in this context that function in NMD similarly to SMG5/7 heterodimers, however, this is not tested or discussed.

- We have quantified the level of overexpression (Extended Data Fig. 4c) and state this now explicitly in the text. We have also tested whether SMG5 and SMG7 can form homodimers when overexpressed (see comment to reviewer #1). However, we did not find any evidence for this.

6. The authors convincingly show in Fig 1 that their SMG7 KO clones have stronger NMD defects than a SMG7 KO (either in their hands or based on previous data by Colombo et al using HeLa cells). This is not unexpected. Moreover, the detailed comparison of transcriptome changes between the SMG7 KO in 293T cells and SMG7 KD in HeLa is comparing apples-to-oranges and distracts from other key findings in the paper.

- We agree with the reviewer on the point that we should not “over-discuss” the differences between SMG7 KD in HeLa cells and the SMG7 KO in 293 cells. However, we believe that the initial analyses (Fig.1e-i and Extended Data Fig. 2b-e) are required to establish that the KO of SMG7 is indeed more effective than the KD. Nevertheless, we have removed superfluous analyses and the corresponding text (e.g. barcode plots).

7. The discussion section of the manuscript and model figure presented in Fig 7 provides a comprehensive overview of NMD events based primarily on published literature and is better suited for a review article. Critically, the authors fail to hone-in on the novel findings gained from their study and how these observations should change how we think about events occurring between (presumably) the NMD machinery and terminating ribosome. There are many provocative findings from this study and the impact of their data and novel roles for SMG5 and SMG7 in eliciting NMD should be the focus of the discussion and better reflected in their model figure(s) and in the manuscript title.

- We completely agree with the reviewer's statement that we should have focused more on our novel findings in the model and discussion section of the previous version of the manuscript. Since we received also concordant feedback from the other reviewers, we completely exchanged our model to better reflect the important implications of our work. Specifically, we show and discuss now three mechanistic hypotheses for the role of SMG5-SMG7 in NMD in more detail.

REVIEWER COMMENTS – second round of revision

Reviewer #1 (Remarks to the Author):

The authors have made substantial changes to the revised manuscript, which has resulted in a robust, clear report. The Turbo-ID study and the modified discussion with three possible hypotheses of how SMG5-7 could act upstream of SMG6 are a great addition. This study will hopefully trigger further research on the obvious open questions of how SMG6 is activated or authenticated and how NMD progresses in cells.

In summary, I have no further questions and support the publication of the manuscript.

- We appreciate the positive assessment by reviewer #1.

Reviewer #2 (Remarks to the Author):

The authors have thoroughly revised their initial version of the manuscript based on the reviewers' comments and the revised version is now much more focused on the main findings, better structured and hence easier and more pleasant to read. Also the revised illustration of the "two factor authentication model" (Fig. 7) has been improved and become clearer. I congratulate the authors to this overall high quality and interesting piece of work, which provides exciting new insights into the still enigmatic mechanism of NMD activation.

My points have all been satisfactorily addressed and I recommend publication of the manuscript in its present form.

(This review is from Oliver Mühlemann)

- We thank Oliver Mühlemann for the positive evaluation of our manuscript.

Reviewer #3 (Remarks to the Author):

Nonsense-mediated mRNA decay relies on “two-factor authentication” by SMG5-SMG7

In their revised manuscript, Boehm et al. address the majority of the concerns raised by this and the other reviewers. The inclusion of important controls, extended analyses, and co-immunoprecipitation and proximity labeling data all strengthen the conclusions and help shed light on the requirement of SMG5-SMG7 for SMG6 activity on NMD substrates. Additionally, they have done a worthy job at rewriting the results and discussion to better highlight the key findings from their data. The manuscript is now much clearer and - with the additional supporting data - provides convincing evidence for the interdependence of SMG5-SMG7 and SMG6 in targeting substrates to NMD.

The only remaining comments relate to both the manuscript title and models they offer to represent their findings. While there is indeed evidence of a hierarchical requirement for NMD factors (and likely significant mRNP remodeling) in targeting of NMD substrates, the concept of two ‘authentication’ steps is not as clear and perhaps misleading to readers, given the lack of full understanding of the events. Likewise, although provocative, evoking this authentication concept in the title does little to describe the major (and important) mechanistic findings presented in the paper. A title such as “Endonucleolytic cleavage of NMD targets by SMG6 relies on SMG5-SMG7” is much more informative to the reader. Finally, presentation of three possible models to explain their findings (Fig. 7) is confusing, given that at least one of these (7a) is not supported by their own data. Presentation of one favored model would be preferred.

- Due to the shortening of the discussion, the 2-factor authentication model is not described in as much detail as in the original version. Furthermore, during the revision of the manuscript, its focus has slightly changed. To reflect these changes, we have revised the title of the manuscript.
- As noted by the reviewer, we presented three possible models in the discussion to explain the activity of SMG5-7 during NMD. Given our own and other's current data, not all of these models are equally likely, so we have decided to describe three different models in writing and present one of them in addition as a figure. We hope that this will further improve the comprehensibility.